

# The dependence of albedo on different factors for refreezing melt ponds in the Arctic

Jialiang Zhu[1], Tao Li[1], Peng Lu[2], Yilin Liu[1], Xiaoyu Wang[3]

[1]College of Oceanic and Atmospheric Sciences, Ocean University of China, Qingdao, 266100, China.
[2]State Key Laboratory of Coastal and Offshore Engineering, Dalian University of Technology, Dalian, 116024, China.
[3]Key Laboratory of Physical Oceanography, MOE, Qingdao, 266100, China.

*Correspondence to*: Tao Li (litaoocean@ouc.edu.cn)

**Abstract.** Sea ice plays an important role in the heat transfer into the Arctic Ocean whereas the presence of melt ponds on sea ice complicates the scenario. In this study, we report a series of observations conducted in the central Arctic during 2012–2020 to investigate the optical and physical properties of refreezing melt ponds. From early August to early September, the albedo of ponds in the Pacific sector increases by 0.0036 $d^{-1}$, which is attributed to the changes on surface state. Based on the typical albedo, the types of melt ponds were categorized as water pond (0.14), water-ice pond (0.20), ice pond (0.25), ice-snow pond (0.39) and snow pond (0.74). Further analysis reveals the capacity of different ratios of spectral albedo on the distinction between snow ponds and unponded ice. In addition, the total albedo of ice ponds decreases with rising pond depth, and the increasing of ice lid thickness reduces the albedo while increases that of ice-snow ponds. Based on the observations, we modified a two-stream radiative transfer model, reducing its remaining error from observation by an order of magnitude. The simulation indicates ice lid thickness as the most important determining factor in the total albedo during the freezing process.

## 1 Introduction

Sea ice plays a vital role in the heat exchange between the ocean and the atmosphere, as well as the solar irradiance penetrating into the ocean, and thus control the radiative forcing in the Arctic Ocean and the world (Hudson, 2011). In the past several decades, the Arctic ecosystem has experienced dramatic landscape changes in sea ice conditions, including the decline in summertime sea ice extent and concentration (Comiso et al., 2008, 2017; Zhao et al., 2018), a reduction in sea ice thickness (Rothrock et al., 2008; Renner et al., 2014), and an extended melt season (Stroeve et al., 2014; Pistone et al., 2014).

In addition, recent studies further report specific changes during the melt season, such as the decrease of snow thickness (Kachimi & Kwok, 2022), the increase of melt pond fraction (Xiong & Ren, 2023), and the change on microstructure of sea ice (Barber et al., 2009, 2012; Meier et al., 2014). The condition has altered the morphologic, thermodynamic, and dynamic properties of the ice cover, which makes difference in the heat budget and mass balance of the Arctic Ocean (Perovich & Polashenski, 2012; Di Biagio et al., 2020), eventually affecting the regional or global climate (Stroeve & Notz, 2018).



The optical properties of sea ice, especially albedo and transmittance, dominate the distribution of incident solar irradiance in sea ice. The sea ice albedo evolves seasonally: in April, the albedo on the surface of Arctic sea ice varies between 0.8 and 0.9 with little spatial variation; by the end of July, the average albedo drops to around 0.4, with a large spatial variation from white bare ice (0.65) to dark melt pond (0.1) (Perovich et al., 2002a). During the melt season, the albedo of bare ice hardly varies due to the surface scattering layer (Light et al., 2015; Smith et al., 2022) so the surface albedo of sea ice in summer is

largely determined by the melt ponds (Light et al., 2022).

Melt ponds are widely observed on sea ice during the Arctic summer: they usually appear in late May and grow wider and deeper in June and July until they refreeze late August and early September (Perovich et al., 2002a). The melt ponds typically reach the highest coverage in early September, with the fraction exceeding 50% for first-year ice (Flocco et al., 2010; Webster et al., 2015) and up to about 30% for multi-year ice (Perovich, 2002b; Webster et al., 2022). Once present,

melt ponds affect sea ice by reducing the regional albedo, increasing the absorption of solar irradiance and accelerating the seasonal melt (Perovich & Polashenski, 2012). In addition, melt ponds mediate the turbulent heat exchange between sea ice and the lower atmosphere (Andreas et al., 2010; Boisvert et al., 2013), as well as the light penetration into the upper ocean, therefore the primary productivity (Nicolaus et al., 2010; Massicotte et al., 2019). Also, Light et al. (2015) reports that the energy entering upper ocean through ponded ice is ~4 times higher than that through unponded ice. Thus, the thermal

characteristics of summer sea ice strongly depends on fraction of melt ponds.

The proper parameterization of melt ponds in numerical simulation is necessary in adequate sea ice forecasting due to its role in regulating the thermodynamic processes of sea ice (Schröeder et al., 2014). The radiative transfer modeling is one of the useful tools to study the importance of melt ponds of different properties: previous model studies show that the albedo of melt ponds is determined by both the pond depth and the substrate ice thickness (Lu et al., 2016) while others develop

respective radiative transfer models for dark and light surface states from several field observations (Malinka et al., 2018).

So far, the seasonal variations of the albedo and fraction of melt ponds of the Arctic sea ice has been well documented. However, few observations and studies have focused on the refreezing of melt ponds (Perovich et al., 2003; Flocco et al., 2015; Anhaus et al., 2021; Light et al., 2008, 2022). Meanwhile, the surface state of varied albedo measurements has been poorly characterized due to the limited dataset (Perovich et al., 2020; Cao et al., 2020). In this study, we aim to fill the gaps

by providing a dataset with detailed albedo-based classification of the surface state, especially with refrozen melt ponds. The description of observation, method, model and surface classification is given in Section 2. The spatial and temporal variation of albedo, as well as the relationship between physical and optical properties are presented in Section 3. Section 4 is some discussion on the distinction of frozen ponds. In Section 5, the main finding is summarized.





## 2 Data and methods

**2.1 In-situ observation**

During Chinese Arctic Scientific Expeditions (CHINARE) from 2012 to 2020, irradiance and properties of melt ponds with different states were measured and recorded at a total of 25 stations on the multi-year ice in the Central Arctic, as shown in Fig. 1 and Table 1. The observation covered the Western Canada Basin, Chukchi Plateau, Mendeleev Ridge, Makarov Basin, Lomonosov Ridge, and Amundsen Basin, most of which conducted in the late summer and early autumn, from August 4th to

September 2nd, under overcast skies. The duration varied from 10 to 20 minutes at short-term ice stations and from 2 to 12 hours at long-term ice stations, with a sampling interval of 1–3 seconds, and the average were recorded as the values of different irradiance. The thickness of ice on the surface of the melt pond (hereinafter referred to as ice lid thickness), the depth of the melt pond, and the ice thickness below the melt pond, i.e., the distance from the bottom of the melt pond to the bottom of the sea ice (hereinafter referred to as substrate ice thickness) were measured and recorded at different locations.

**Table 1: Station information of observations during CHINARE 2012–2020.**

| Year | Station ID | Date | Longitude | Latitude | Melt ponds | Ice lid thickness | Pond depth | Substrate ice thickness |
|------|-----------|------|-----------|----------|-----------|------------------|-----------|------------------------|
| 2012 | IC1201 | 8.29 | 120°23.947'E | 86°48.023'N | 4 | | √ | |
| | IC1202 | 8.30 | 123°24.627'E | 87°39.603'N | 5 | | √ | |
| | IC1203 | 8.31 | 120°14.885'E | 86°36.910'N | 6 | | √ | |
| | IC1204 | 9.01 | 145°14.847'E | 84°59.976'N | 3 | | √ | |
| | IC1205 | 9.02 | 158°46.952'E | 84°05.584'N | 7 | | √ | |
| | IC1206 | 9.02 | 161°41.588'E | 83°37.646'N | 6 | √ | √ | |
| 2014 | IC1401 | 8.10 | 151°06.283'W | 76°41.917'N | 5 | √ | √ | |
| | IC1402 | 8.12 | 154°35.342'W | 77°10.897'N | 3 | | √ | |
| | IC1403 | 8.13 | 163°08.102'W | 77°29.293' N | 4 | √ | √ | |
| | IC1405 | 8.16 | 158°37.558'W | 79°55.930'N | 4 | | √ | |
| | IC1408 | 8.28 | 149°21.510'W | 78°48.362'N | 3 | √ | √ | |
| 2016 | IC1601 | 8.04 | 169°09.516'W | 78°59.197'N | 3 | | | |
| | IC1602 | 8.05 | 169°05.405'W | 80°05.602'N | 3 | | | |
| | IC1603 | 8.06 | 167°39.673'W | 81°33.087'N | 4 | | | |
| 2018 | IC1802 | 8.12 | 168°05.950'W | 79°55.840'N | 2 | √ | √ | √ |
| | IC1804 | 8.14 | 168°11.170'W | 82°19.100'N | 2 | √ | √ | √ |
| | IC1805 | 8.15 | 167°21.380'W | 82°37.550'N | 2 | √ | √ | √ |
| | IC1808 | 8.23 | 162°10.140'W | 84°34.690'N | 3 | √ | √ | √ |
| | IC1809 | 8.24 | 156°06.230'W | 84°24.440'N | 5 | √ | √ | √ |
| 2020 | IC2004 | 8.19–8.21 | 161°13.599'W | 86°02.747'N | 6 | | | |
| | IC2005 | 8.23 | 170°28.000'W | 85°37.000'N | 1 | | | |
| | IC2006 | 8.25 | 174°42.175'E | 85°12.060'N | 1 | | | |



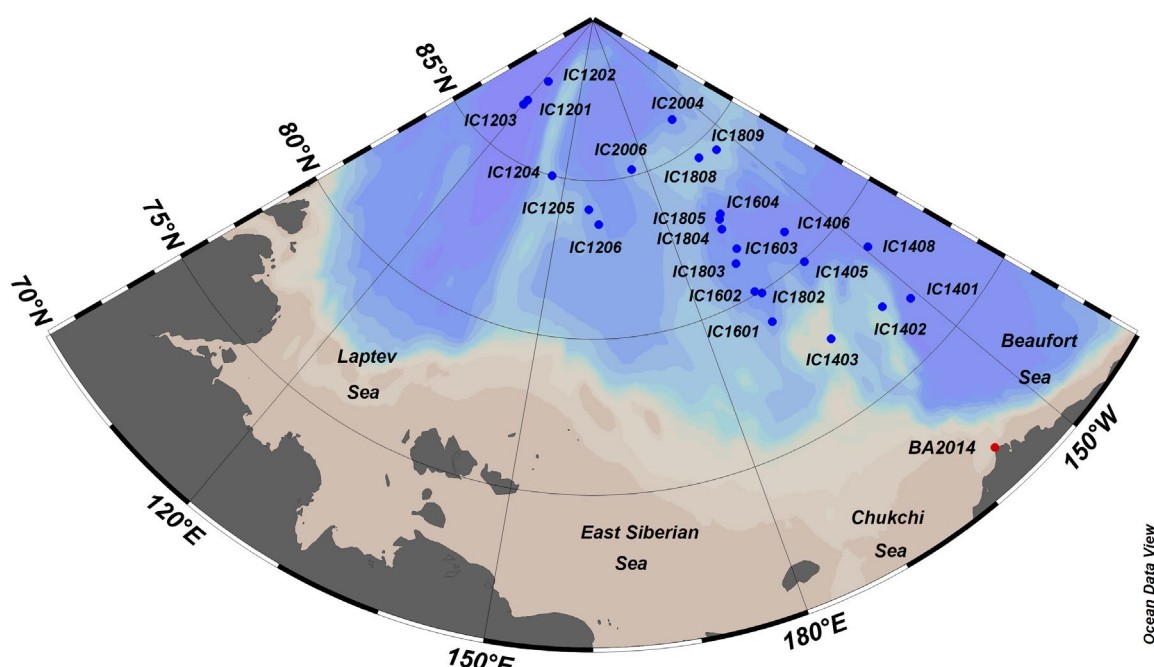

**Figure 1: Station distribution of observations. Blue dots represent the locations of ice stations during CHINARE 2012–2020 and red dot represents the location of observation in 2014, Barrow.**

Two types of radiometers were used to obtain the incident and reflected irradiance. The CNR4 radiometers (KIPP&ZONEN,
Germany) were used in 2012, 2014, and 2016 (Fig. 2a), which measure the incident and reflected integrated irradiance in the
wavelength range from 300 to 2800 nm. In this way, the total albedo was derived from the irradiance. The Ramses ACC-VIS
hyperspectral radiometers (TriOS, Germany) were used in 2018 and 2020 (Fig. 2b), covering wavelength ranging from 320
to 950 nm with 195 channels at a resolution of ~3 nm (Nicolaus et al., 2010). Both instruments were used simultaneously
during the sea ice observation in Barrow, Alaska, 2014 (see Fig. 1). The dataset acquired in Barrow was applied in the
calibration (see Section 2.3) to unify the band of integrated irradiance measured by different instruments before further
analysis.

The instruments were mounted at the end of an extension pole which was fixed on a tripod and counterweighted to be
parallel with the surface (Fig. 2a, 2b). This ensured that the sensor for upward irradiance was about 0.8 m above the melt
pond, thus reducing the interference from the edges. The surface condition and physical properties (ice lid thickness, pond
depth and substrate ice thickness) were measured. Note that the depth of all melt ponds with a closed bottom were measured,
while measurements of the ice lid thickness were performed on selected locations. In addition, the substrate ice thickness
was measured in CHINARE 2018.



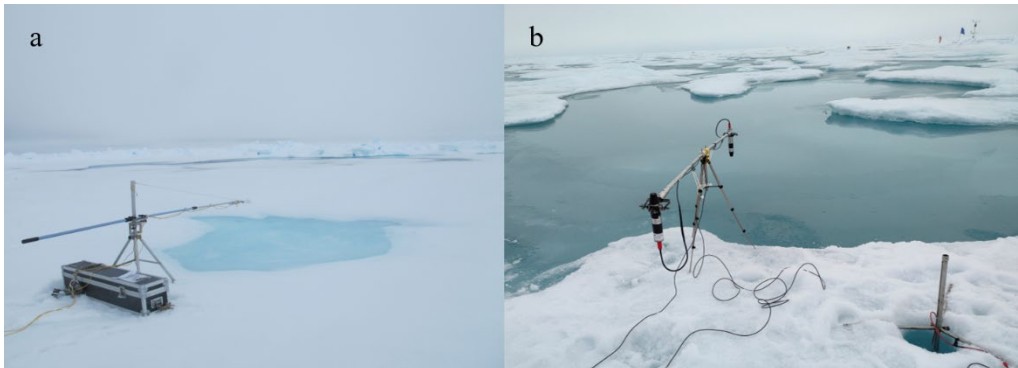

**Figure 2: Radiometer setup for albedo observations in CHINARE. (a) CNR4 (b) Ramses ACC-VIS.**

The albedo of sea ice and melt pond was derived from the measured irradiance after the observation. The spectral albedo is defined as

$$\alpha(\lambda) = \frac{F_u(0, \lambda)}{F_d(0, \lambda)},$$

where $F_u(0, \lambda)$ is the plane upward irradiance, i.e. the reflected irradiance, and $F_d(0, \lambda)$ is the plane downward irradiance, i.e. the incident irradiance (0 represents the sea ice surface). Similarly, the total albedo can be defined as


$$\alpha_t = \frac{\int \alpha(\lambda) F_d(0, \lambda) d\lambda}{\int F_d(0, \lambda) d\lambda}$$

In general, the total albedo in cloudy skies is greater than the albedo in clear skies by 8–12% higher (Perovich, 1996). And it should be noted that observations used in this study were measured with a cloudy sky, since data from observations under clear skies are excluded before analysis.

**2.2 Radiative transfer model**

To further quantify the effects of various factors on the albedo of the melt pond, a two-stream radiative transfer model (Fig. 3) developed by Lu et al. (2016) was modified to simulate the optical properties of the melt pond. It should be note that here we focus more on the improvement with proper parameterization and observed range of different properties, so a relatively simple model is selected. The four-layer model is schematically shown as follows.



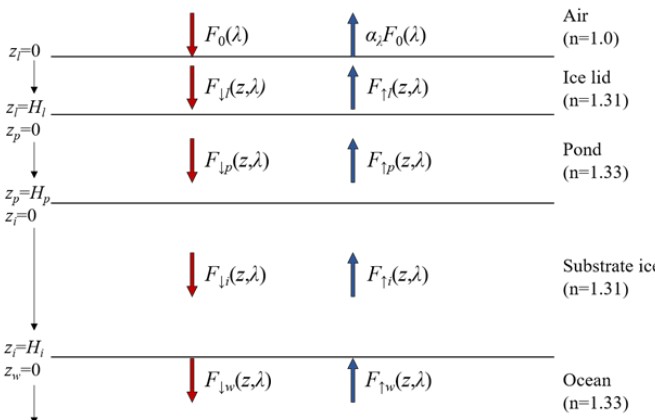

**Figure 3: Schematic graph of the two-stream radiative transfer model for the melt pond.** $F_0(\lambda)$ is the incident solar irradiance, $F_\uparrow(z, \lambda)$ and $F_\downarrow(z, \lambda)$ represent the upward and downward irradiance, $H$ is the thickness, with different subscripts $p$, $i$ and $w$ represent the pond water, the substrate ice and the ocean, respectively. $n$ is the refractive index of different layers.

In this model, sea ice is treated as Isotropic, and the inherent optical properties of each layer were determined by the wavelength-dependent absorption coefficient $k_\lambda$ and the scattering coefficient $\sigma_\lambda$, a constant independent of wavelength. Under the assumption of diffuse incident solar irradiance and isotropic scattering, the upward and downward irradiance of each layer are governed by two coupled first-order differential equations:

$$\begin{cases} dF^\downarrow(z,\lambda) = -k_\lambda F^\downarrow(z,\lambda)dz - \sigma_\lambda F^\downarrow(z,\lambda) + \sigma_\lambda F^\uparrow(z,\lambda)dz \\ dF^\uparrow(z,\lambda) = k_\lambda F^\uparrow(z,\lambda)dz + \sigma_\lambda F^\uparrow(z,\lambda) - \sigma_\lambda F^\downarrow(z,\lambda)dz \end{cases}$$

where $z$ increases downward. The general solution is

$$\begin{cases} F^\downarrow(z,\lambda) = A(1-\mu_\lambda)\exp(\kappa_\lambda z) + B(1+\mu_\lambda)\exp(-\kappa_\lambda z) \\ F^\uparrow(z,\lambda) = A(1+\mu_\lambda)\exp(\kappa_\lambda z) + B(1-\mu_\lambda)\exp(-\kappa_\lambda z) \end{cases}$$

where $\mu_\lambda = \sqrt{k_\lambda/(k_\lambda + 2\sigma_\lambda)}$ is the absorption strength (0 for purely scattering medium and 1 for purely absorbing medium), $\kappa_\lambda = \sqrt{k_\lambda(k_\lambda + 2\sigma_\lambda)}$ represents the attenuation coefficient as defined in Perovich (1990). $A$ and $B$ are constants determined by the boundary conditions. The band of incident solar irradiance $F_0(\lambda)$ in this model is set to 400–900 nm, as 70–80% of the solar irradiance energy reaching the Earth's surface distributes in this range (Liou, 2002). Since the refractive indices of ice and water are similar, the Fresnel reflection coefficient between water and ice can be neglected ($R_2 = 0$). The reflection at the air-water interface is taken as $R_1=0.05$ for the diffuse sky, according to Perovich et al. (1990). In addition, the scattering of solar irradiance in the pond water and the ocean has been neglected, so $F_{\uparrow w}(z, \lambda)$ is assumed to be 0 in this model.

The coefficients of the inherent optical properties for the substrate ice are modified based on the field record to ensure the simulation to be consistent with the observation. Wang et al. (2020) reports that the volume of bubbles inside the sea ice decreases while the volume of brine increases with the increasing of depth, which lead to the inhomogeneous optical properties of the bottom ice of the melt pond. Here a combination of attenuation coefficient for white ice interior and pure ice in Perovich et al. (1990) is adopted, instead of that for pure ice used in original settings. According to Perovich et al.




(1990), the scattering coefficient of white ice interior is 2.5 m$^{-1}$, while Light et al. (2015) argue that the scattering coefficient of substrate ice varies between 10 and 22 m$^{-1}$, and a value of 13 is taken in the multi-layer model (Light et al., 2008). In this study, as most of the melt ponds observed are dark ponds and the resulted high scattering coefficient is one order of

magnitude higher than the observed, so the scattering coefficient of substrate ice is set to 2 m$^{-1}$, consistent with Malinka et al (2018) and Katlein et al. (2015). Besides, the incident irradiance, ice lid thickness, pond depth and substrate ice thickness are all adopted from the in-situ observation.

## 2.3 Surface classification and calibration

The sea ice surface is heterogeneous and often a mixed presence with water, snow and ice, therefore with varied albedo

conditions. As a result, the melt pond observed are classified into five categories, namely water pond, water-ice pond, ice pond, ice-snow pond, and snow pond. Typical states of the five types and the unponded ice are shown in Fig. 4. Specifically, the water pond (Fig. 4a) is defined as a melt pond with a surface of liquid water that has not yet started refreezing; the surface of a water-ice pond (Fig. 4b) is partly frozen yet not covered completely by the ice lid; the ice pond is when a melt pond is entirely frozen on its surface with neither water nor snow (Fig. 4c); the ice-snow pond is defined as a melt pond with

ice lid partially covered by snow (Fig. 4d); the snow pond (Fig. 4e) is defined as a melt pond with its surface frozen and totally covered by snow , which makes it hard to distinguish from the snow-covered unponded sea ice (Fig. 4f). And it should be noted that all types of melt ponds mentioned above, except for water pond, have liquid water underneath their lids.

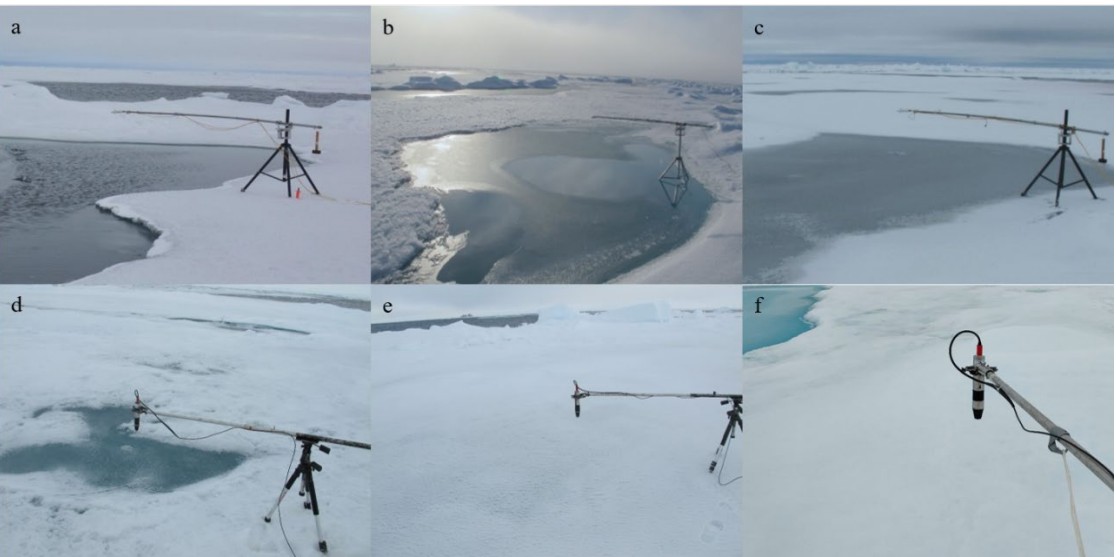

**Figure 4: Typical surface states of (a) water pond, (b) water-ice pond, (c) ice pond, (d) ice-snow pond, (e) snow pond and (f) snow-**
**covered unponded ice.**

As described in Section 2.1, two different instruments were used in the field observations during CHINARE 2012–2020. As a result, the broadband values of irradiance were calculated by integrating over 300–2800 nm in 2012–2016 while over 320–



950 nm in 2018–2020. So, the integrated irradiance obtained from CNR4 in 2012–2016 needs to be calibrated into 320–950 nm to make reasonable comparison in following sections. The coefficients for calibration are derived from the ratio of integrated values measured simultaneously by CNR4 and Ramses ACC-VIS in the observation at Barrow, 2014 (Fig. 5a, 5b). Comparisons of irradiance obtained from the two instruments before and after calibration are shown in Fig. 5c and Fig. 5d.

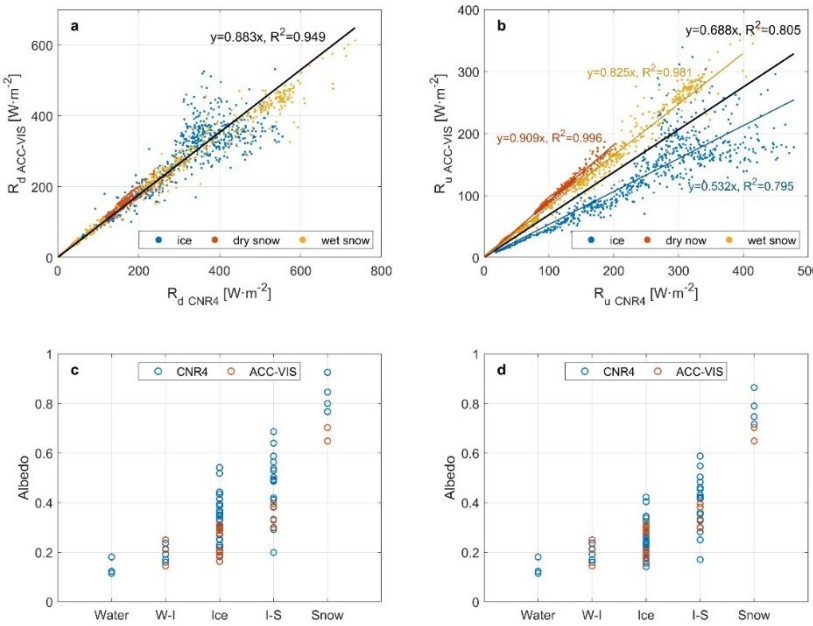

**Figure 5: (a) Downward integrated irradiance, (b) upward integrated irradiance, (c) raw albedo and (d) calibrated albedo measured by CNR4 and Ramses ACC-VIS for different surface states and types of ponds. The solid lines in panels (a) and (b) are the linear fitting for ice (blue), dry snow (red), wet snow (yellow) and overall (black), with the annotations showing the fitting coefficients. The abbreviation W-I represents water-ice pond and I-S represents ice-snow pond.**

In Barrow, 2014, CNR4 and Ramses ACC-VIS were placed at the same location on the sea ice from May 10th to May 13th to conduct consecutive observations (Zhu et al., 2021) with variable local meteorological states. Despite the sea ice surface varied between ice, dry snow and wet snow due to precipitation, the downward irradiance ($R_d$) and upward irradiance ($R_u$) integrated over different bands showed prominent correlation. The ratio of $R_d$ measured by Ramses ACC-VIS to that of CNR4 is 0.883, which means the integrated value of irradiance over 320–950 nm is 88.3% of that over 300–2500 nm, and remain unchanged during the observation (Fig. 5a). In contrast, the ratio of reflected irradiance varied with different surface states. As shown in Fig. 5b, $R_u$ measured by Ramses ACC-VIS reached 0.532, 0.825 and 0.909 of that measured by CNR4 at the surface state of ice, wet snow and dry snow, respectively. For calculation, 0.825 was taken as the calibration coefficient for snow ponds, 0.532 was taken for ice ponds, while an average value of 0.678 was used for the ice-snow ponds. $R_d$ and $R_u$ measured by CNR4 are calibrated and then used to derive total albedo over 320–950 nm, same with the range of Ramses ACC-VIS. As a result, the calibration reduces the median deviation of CNR4 measurements from 0.2 to 0.06 (Fig. 5c and 5d).



# 3 Results

## 3.1 Spatio-temporal characteristics

As described in Section 2.1, the observations were conducted between August 4th and September 2nd, when the melt ponds start the fall freezing (Perovich & Polashenski, 2012). The albedo of melt ponds varied spatially and temporally and was dependent on surface states from water to snow. The temporal variation of the albedo is presented in Fig. 6a. From day 216 to day 235, the albedo of melt ponds is 0.28±0.12, except for several snow ponds observed on day 225, which has a maximum albedo of 0.69. In day 235–245, the albedo increases significantly to 0.32±0.11 (0.7 at maximum) in the first 5 days, and then to 0.36±0.21 (0.86 at maximum) in the last 5 days. During this month, the five-day average total albedo of the melt ponds gradually increased by ~0.1 from 0.27 to 0.36 at a rate of 0.0036 d$^{-1}$. The observed rate is smaller than that observed in SHEBA, 1998 (0.008 d$^{-1}$, Perovich et al., 2007) and that in Arctic Transpolar Drift, 2007 (0.006 d$^{-1}$, Nicolaus et al., 2010), indicating a possible slowdown on the fall freezeup of melt ponds. However, this assumption is not robust since the observations were conducted in different years, and the difference between each sector of Arctic was not included. A rate of about 0.01 d$^{-1}$ was observed for a refreezing pond in late August during MOSAiC expedition (Light et al., 2022), and the albedo rose over 0.1 as the surface partly covered by snow.

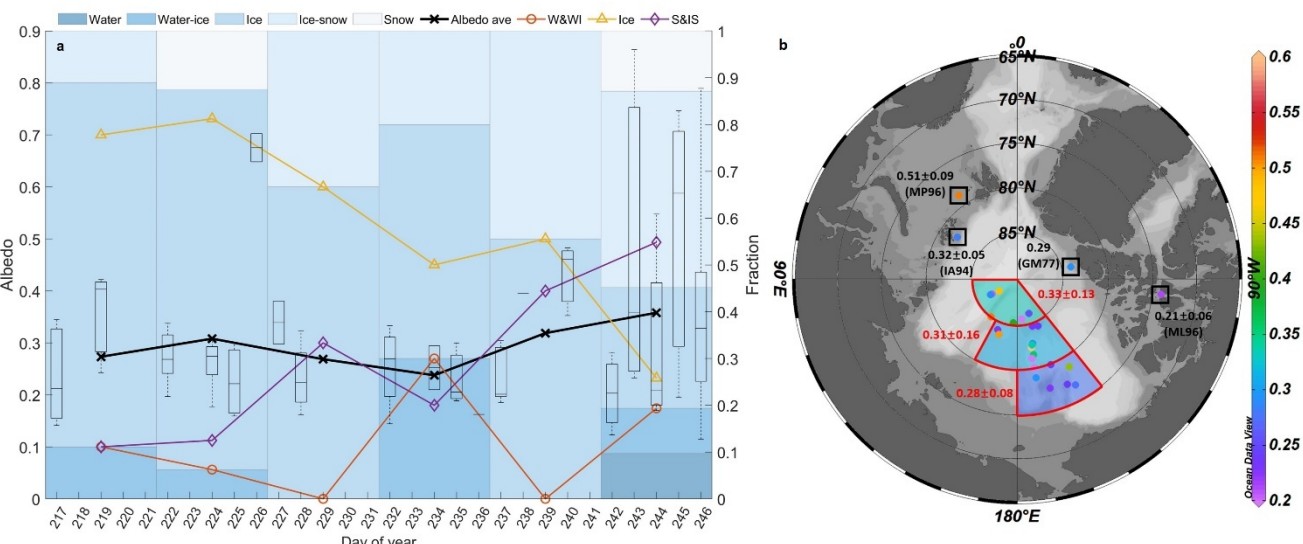

**Figure 6: The temporal (a) and spatial (b) distribution of total albedo of melt ponds. In subplot (a), the boxplot represents the albedo of melt ponds observed in certain DOY (day of year), bars represent 5-day average proportions (axis at the right side) of each type of pond, the black solid line is the 5-day average of albedo, dashed lines represent 5-day average of proportions (axis at the right side) of water & water-ice (red) pond, ice (yellow) pond, ice-snow & snow (purple) pond, respectively.**

The increase of albedo appears along with the variation on the proportion of different surface states. As shown in Fig. 6a, 80% of the melt ponds were ice ponds in early August while less than 15% were ice-snow ponds. However, the fraction of ice ponds decreased to 30% by early September with more than 50% covered by snow. Due the presence of the snow cover, the



average albedo of all melt ponds increased. Our observation shows that the albedo of the melt ponds was generally low in early August, but an increased proportion of snow-covered ponds occurs from mid-August onwards, and thus the snow cover gradually became a determining factor for the albedo of melt ponds. The result also indicates that since melt ponds are in different states at a certain moment, the change in the proportion of surface states leads to the overall increase in the albedo of the melt ponds during fall freezeup.


The spatial distribution of melt pond albedo is shown in Fig. 6b. Overall, the total albedo of the melt ponds on Arctic sea ice increased with latitude with a mean albedo of 0.28 between 75°N and 80°N, 0.30 in 80°N–85°N, and exceeding 0.33 at the north of 85°N. This pattern is related to the fact that the incident solar irradiance in the Arctic region decreases with increasing latitude, which makes difference on the surface properties. Melt ponds at higher latitudes receive less energy through shortwave irradiance, thus refreeze faster and the albedo is higher than those at south. But it should also be noted that all stations northerly than 85°N were measured in late August so the higher albedo in this area contains the effect of temporal factors. In addition, the standard deviation of melt pond albedo is highest for the area 80°N–85°N, which was caused by the more surface states as well as the more wide-ranged sampling dates. This is consistent with the albedo pattern shown in Fig. 6a. Snow cover on the melt ponds is affected by the difference on physical properties such as topography of ponds and surrounding sea ice (Perovich et al., 2003; Anhaus et al., 2021), the earlier freeze onset and faster freezing process makes melt ponds at higher latitudes prone to develop a surface with more snow, contributing to a higher albedo than other regions.



## 3.2 Effects of the surface state

The total albedo over 320–950 nm of the various types of melt ponds is presented in Fig. 7. In August, the albedo of melt ponds in the central Arctic Ocean is mostly in the range of 0.2–0.5 (Fig. 7a), and the proportions of melt ponds with albedo of 0.15–0.2, 0.2–0.25 and 0.25–0.3 are 18.5%, 20.1%, and 14.8%, respectively. In comparison, albedo of melt ponds was seldomly found with values greater than 0.5 (11.1%). There were only few observations for water ponds due to the timing when melt ponds began to freeze (Perovich et al., 2007), and the total albedo ranges from 0.11 to 0.18, with an average value of 0.14. Water ponds are those at the beginning of refreezing, so the albedo is a little higher than the open water (~0.08) due to the existence of substrate ice (Fig. 7b). This result is lower than previously reported values for melt ponds on multi-year ice in SHEBA (0.4) (Perovich et al., 2002a), but close to those on seasonal ice (0.2) (Perovich et al., 2012) and dark ponds (0.12) (Malinka et al., 2018), as well as the dark ponds (0.12–0.25) observed during MOSAiC (Light et al., 2022). The albedo mean of water-ice ponds reaches 0.20, with an overall variation less than 0.05 compared to the water ponds, indicating that the newly formed ice lid on the surface has little effect on the albedo during the early stages of refreezing.



Most melt pond observations were conducted in mid to late August when melt ponds have been refreezing for half a month. The albedo of ice ponds ranges from 0.15–0.42, mostly between 0.2 and 0.3, with a mean value of 0.25, while the albedo of ice-snow ponds varies between 0.17 to 0.58, most of which between 0.32–0.45 (average: 0.39). The standard deviation of ice-snow ponds is 0.1, the highest among all five states, due to the heterogenous characteristic of the snow cover. Our results






show that the presence of partial snow cover exerts considerable yet limited influences on the albedo of melt ponds that is

much lower than the typical value of for snow-covered ice (0.8) and of bare ice (0.7) (Perovich et al., 1990; 2002a), the

situation changes significantly when it is completely covered with snow. The albedo of snow ponds is 0.74, which is closer

to that of typical snow-covered sea ice (~0.8) rather than other types of ponds, making it hard to distinguish from the

surrounding unponded sea ice. Furthermore, the albedo of snow ponds is nearly three times the albedo of ice ponds while the

albedo of ice-snow ponds generally exceeds that of ice ponds. This implies the crucial effects of snow cover on the total

albedo of a melt pond.

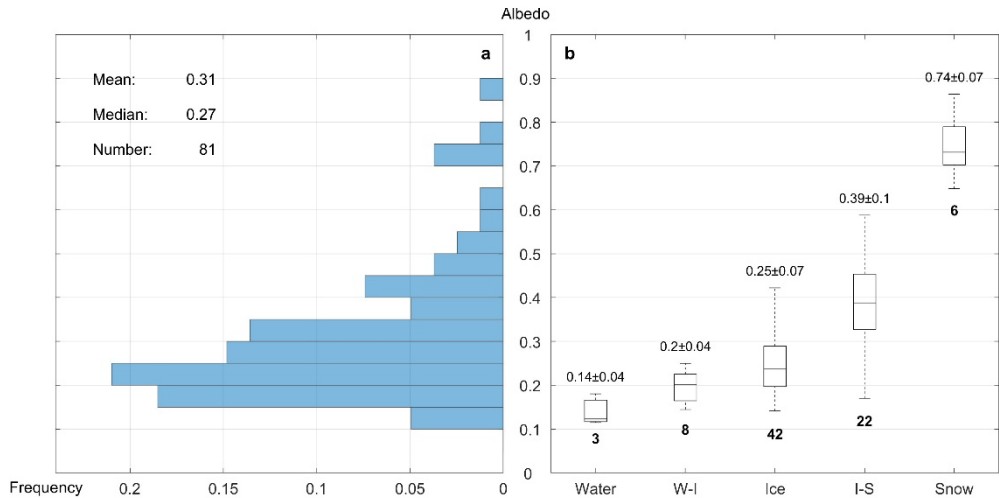

**Figure 7: Albedo distribution of all melt ponds observed with (a) frequency and (b) types of surface states, where W-I represents water-ice pond and I-S represents ice-snow pond. The total numbers of ponds sampled for each type are annotated by the bold number under the boxplot in panel (b).**

Due to the large differences in the inherent optical properties of snow, ice, and water (Perovich et al., 1990), changes in the

surface state of sea ice and melt pond have effects on both the integrated value and the spectral distribution of albedo, as

plotted in Fig. 8. Since less energy of solar radiation concentrates in short wavelength and causes noises in 320–350 nm, the

valid data for spectral distribution in this section start with 350 nm. The spectral albedo of ice ponds varies between 0.10 and

0.34 at different wavelengths, reaching its maximum at 480 nm. Similar trends of spectral distribution also appear on water-

ice ponds and ice-snow ponds, whereas the three states mainly differed in the overall magnitude, with a mean difference of

0.05 between water-ice and ice ponds, and 0.06 between ice and ice-snow ponds. There is a slightly difference on location of

the peak albedo among these three types of ponds, which shifts from 452 nm for water-ice ponds to 470 nm for ice-snow

ponds. The maximum value of those ponds does not exceed 0.4 though the presence of snow on part of the surface, which is

consistent with dark ponds observed in previous studies (Light et al., 2015; Malinka et al., 2018). However, the spectral

albedo of snow ponds is much higher than the other ponds, with the lowest value (~0.5) measured around 350 nm while the

measurements generally exceed 0.65 in 500–600 nm. In addition, the spectral albedo at 900 nm for snow ponds is 0.54,



which is nearly one order of magnitude higher than that of water-ice ponds. This also suggests that the albedo characteristic of melt ponds is still dominated by ice and water during the earlier stage of freezeup before the entire surface gets covered by snow.

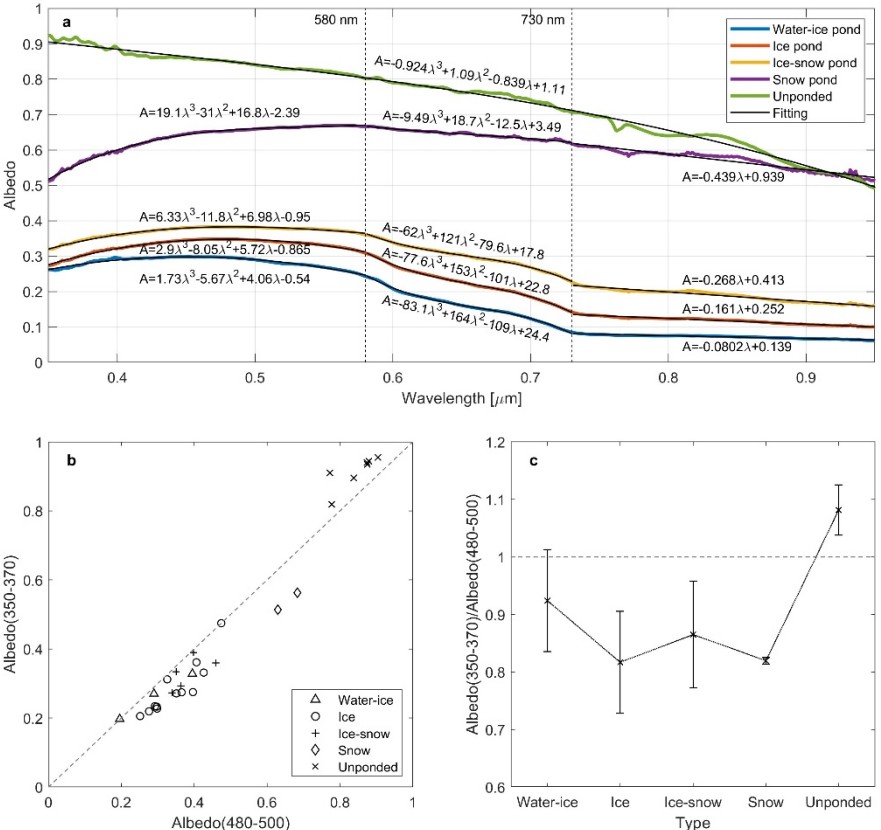

**Figure 8: (a) Spectral distribution of different types of melt ponds and unponded snow-covered ice in 350–950 nm, and the ratio of spectral albedo between band 350–370 nm and band 480–500 nm for (b) all ponds and (c) different types. The non-fitted lines in panel (a) are averages of the ponds observed for every individual type (3 for water-ice pond, 12 for ice pond, 5 for ice-snow pond and 2 for snow pond) except water pond, which was not observed by spectroradiometer in 2018 and 2020.**

The gradient of spectral albedo from 350 to 500 nm increases gradually with the deepening of refreezing process, that is, the change of water-ice-snow. Specifically, the albedo difference between 360 to 490 nm is 0.05 for water-ice ponds, 0.08 for ice and ice-snow ponds, while up to 0.23 for snow ponds. Besides, the wavelength where peak value appears shifts from ~450 nm for water-ice ponds to ~550 nm for snow ponds, showing a trend of increasing as the pond evolves, consistent with that observed in Light et al., 2008. The albedo of all ponds drops as wavelength exceeding 550 nm, while the gradient decreases and the value increases as the surface state changes from water-ice to ice-snow. Even though the snow ponds show a much higher albedo than the other types of ponds, there is still an increasing trend between 350 and 550 nm, which is the distinctive characteristic of melt ponds during fall freezeup (Nicolaus et al., 2010; Malinka et al., 2018; Light et al., 2008,



2022). In contrast, the spectral albedo of unponded sea ice decreases almost at all bands, from the maximum value of 0.92 at 350 nm to less than 0.8 at 550 nm. At 950 nm, the albedo of unponded sea ice reaches the minimum of 0.5, a similar value with snow pond. Besides, for unponded ice, the gradient of spectral albedo over 550 nm is larger than that of snow ponds. The result indicates that it is possible to distinguish snow ponds from unponded ice by spectral albedo before the pond completely refreezes. Furthermore, it can be implied that the spectral distribution of radiation under a visually similar snow-covered surface is not spatially homogeneous due to the existence of snow ponds, which may lead to discrepancy on the energy balance and affect the overall rate of sea ice refreezing.

The spectral albedo of ponds with different states is fitted as shown in Fig. 8a. The fitting is performed in segments based on the corresponding spectral distribution pattern separated by 580 nm and 730 nm. The variation is slow in the first and third segments, while rapid in the second segment. The exception is unponded ice that no segment is performed as its albedo shows no abrupt change on gradients, so a polynomial fit is applied for the full band. It should be noted that the unit of wavelength is μm. The results show regular variations with the refreezing of ponds, which is reflected on the increasing on coefficients in the first and third segments, and decreasing in the second segment. Besides, for unponded ice and snow ponds, there is an obvious nonlinear pattern between 730 and 950 nm, while a slight similar pattern exists on the same band for the rest types which enhances as snow cover accumulates, so it can be considered as the influence of the optical properties of snow, which is consistent with observations in Malinka et al. (2016).

Based on the albedo characteristics described above, the spectral albedo at 360 nm (shortest valid wavelength in this observation) and 490 nm (wavelength where most albedo maximum appear) is chosen as the indicator of melt pond (Fig. 8b). For ponded ice, spectral albedo at 490 nm is higher than that at 360 nm, regardless of the surface states, while for unponded ice the former one is always smaller than the latter. Therefore, it can be assumed that for ponded sea ice the ratio between albedo at 360 nm and albedo at 490 nm is less than 1 and unponded sea ice shows the opposite. In order to avoid uncertainty (e.g. interrupt of noise at ultraviolet) of single wavelength, the integral value within ±10 nm is used as the value of certain wavelength, which means we define the typical albedo at 360 nm as the integral albedo within 350–370 nm (hereinafter referred to as $\alpha_{360}$). Similarly, $\alpha_{490}$ represents the typical albedo at 490 nm and is defined as the integral albedo within 480–500 nm. As shown in Fig. 8c, the ratio $\alpha_{360}/\alpha_{490}$ decreases as the ponds freeze up, with an average from 0.7 to 0.9, in contrast to the mean value of 1.1 for unponded ice. It should also be noted that the ratio $\alpha_{360}/\alpha_{490}$ and $\alpha_{412}/\alpha_{667}$ in this study are developed based on the observation conducted in early freezeup, so limitation exists when adopting to another time or region, e.g. mid-September when the albedo increases along with the wavelength in ultraviolet (Light et al., 2022). And we have to admit that some uncertainty remains in this result as the number of samples is not big enough, but this may still help on the further development of melt pond detection algorithm.

**3.3 Dependence on pond depth and ice thickness**

While range of the total albedo is basically determined by the surface state, the depth of pond water would make a difference for melt ponds sharing the same state. In total, the melt pond depth was measured on 50 of the 81 observations and the





thickness of the ice lids was measured on ice ponds (17 of 50). In August, there were almost equal numbers of melt ponds of different depth (Fig. 9a). The depth distribution did not follow a specific surface state that a large range of pond depth was found for all surface states except the water ponds with only two observations (Fig. 9b). The majority of ice ponds falls in the range of 0.16–0.32 m, smaller than the rest which ranges between 0.3 and 0.5 m. Besides, a correlation coefficient of 0.12 is found between the total albedo and the pond depth for ponds with snow cover, indicating that the albedo of ice-snow and snow ponds is independent of the depth.

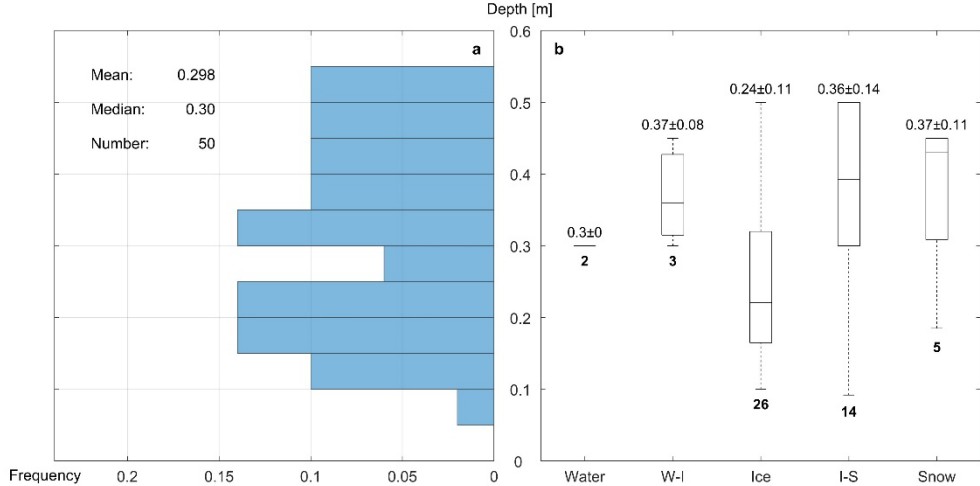

**Figure 9: Distribution of the depth of melt ponds with (a) frequency and (b) types of surface states, where W-I represents water-ice pond and I-S represents ice-snow pond. The total numbers of ponds sampled for each type are annotated by the bold number under the boxplot in panel (b).**

For melt ponds without snow cover, the depth dominates the albedo even if the surface has frozen up. Based on the mechanism of radiative transferring in ice pond, an exponential fitting was adopted on the decreasing albedo and increasing pond depth (Fig. 10a). The fitting result shows that the albedo decreases rapidly when the depth is less than 0.1 m, reaching 0.32 at 0.1 m, less than half of 0.75, the fitted value of unponded ice. The albedo decrease with depth slows down when the depth exceeds 0.1 m, with an albedo difference of only 0.05 between the depth of 0.1 and 0.5 m. Compared with the result of previous studies, the albedo decreases faster at small depth while slower at large depth. The fitting result is close to that of ML96 and SP07 due to the similar bands integrated, and differs with that in EC93 which uses a shorter wavelength. Likewise, it is also attributed to the distribution of spectral albedo that the mean of total albedo in MP96 and IL17 reaches 0.51 and 0.44, respectively, which is nearly twice the average in this study (0.24).





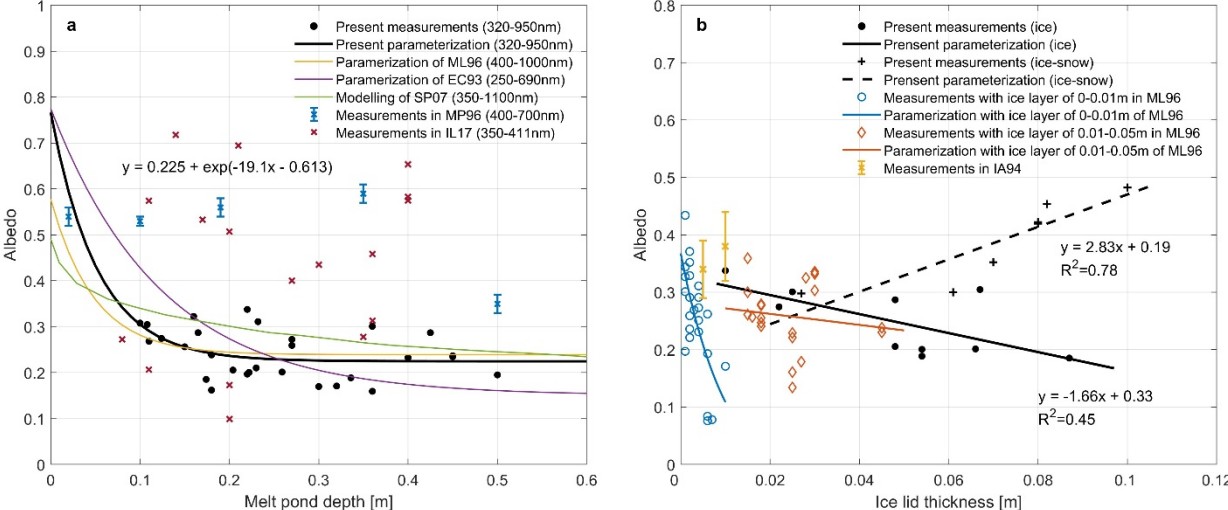

**Figure 10: The fitting results for (a) albedo and melt pond depth, (b) albedo and ice lid thickness. In panel (a), solid lines represent fitting in different studies and dots are the measured albedo, where the acronym is as follows: ML96 – Morassutti & Ledrew, 1996; EC93 – Ebert & Curry, 1993; SP07 – Skyllingstad & Paulson, 2007; MP96 – Makshtas & Podgorny, 1996; IL17 – Istomina et al., 2017. In panel (b), the lines represent fitting in different studies and dots are the measured albedo, where the acronym is as follows: ML96 – Morassutti & Ledrew, 1996; IA94 – Ivanov & Alexadrov, 1994.**

For ice ponds, the relationship between the total albedo and the ice lid thickness is investigated as plotted in Fig. 10b. According to Morassutti and Ledrew, (1996), the behavior of thickness on albedo differs a lot between newly formed ice lid and thicker ice lid (> 0.01 m). In our study, all ice lids exceed 0.01 m and therefore a linear relationship is found, showing that albedo decreases with the increase of ice lid thickness, which is consistent with that in ML96 for ice lid between 0.01 and 0.05 m. However, linear regression for ice-snow ponds (dash line in Fig. 10b) displays a considerable positive correlation between the albedo and the ice lid thickness, suggesting snow accumulation on the surface may cause an albedo variation of 0.23 while thickness grows from 2 to 10 cm. The ratio of the number of ice ponds to ice-snow ponds is about 2:1 in August, and the ratio will decrease in September and October with lower temperature on the snow cover. In this case, the overall albedo of all ponds will rise with the growth of ice lid, reducing the energy obtained from solar irradiance and accelerating the freezeup.

### 3.4 Contribution of the substrate ice for ice ponds

A total of 7 ice ponds were measured for the three physical properties, including ice lid thickness, pond depth and substrate ice thickness. Based on the observed properties, a radiative transfer model is applied to further investigate and parameterize their effects on the spectral albedo. The simulation result is plotted together with the observed values as in Fig. 11, where subplot (a) presents all results and subplots (b)–(h) show results of each pond separately. The maximum value of spectral albedo for pure ice appears at the wavelength of 450 nm, and the peak usually moves towards 550 nm for most sea ice, with




its value dropping by 5–10%, that is due to the interference of particulate matter (Perovich et al., 1998; Light et al., 2008). In this study, the concentration is set to 10 g m⁻² referring to that in Light et al. (1998).

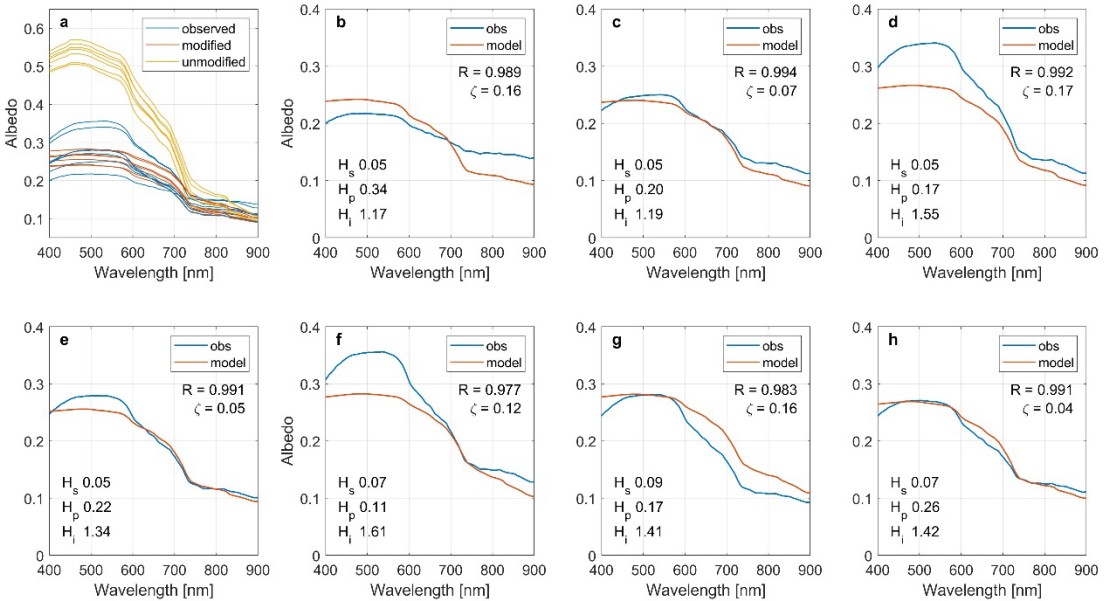

**Figure 11: The observed and simulated values of spectral albedo for melt ponds, where (a) shows the observed albedo, and the result of model with and without modification of all ponds. Panels (b)–(h) present the result of each pond separately, where $R$ represents correlation coefficient, $\zeta$ represents relative error, $H_s$ is the ice lid thickness, $H_p$ is the melt pond depth and $H_i$ is the substrate ice thickness.**

As shown in Fig. 11, the spectral albedo increases in the band 400–550 nm, decreases rapidly in 580–720 nm and slowly in 720–900 nm. A good simulation is found in 450–900 nm and not in 400–450 nm where the rapid rise is not reproduced. Besides, the simulation shows that spectral albedo is affected by all the ice lid thickness, pond depth and substrate ice thickness, yet within a small range (Fig. 11a). In the simulation, discrepancy among all ponds is reflected on the overall magnitude (~0.05 for the albedo maximum) instead of the spectral distribution or gradient which have little variation. However, the distribution interval of spectral albedo for different ponds ranges from 0.09 (Fig. 11b) to 0.23 (Fig. 11f), while the difference on the overall magnitude reaches 0.15. Combined with the physical properties of those ponds, it can be inferred that the substrate ice thickness, compared with the ice lid thickness and melt pond depth, dominates the spectral albedo, especially the spectral distribution which is not fully represented in the simulation. In the melt pond with the lowest substrate ice thickness (Fig. 11b), the observed gradient is obviously smaller than the simulation, showing a difference 0.06 at 550 nm. Meanwhile, for the melt ponds with the largest substrate ice thickness (Fig. 11d and Fig. 11f), the gradients of observation become larger than that of simulation for ponds, leading to an underestimated albedo maximum. The result shows that the importance of substrate ice thickness is not well represented in the simulation but it is a crucial factor of albedo (Lu et al., 2018; Light et al., 2022), which is more important than the role of ice lid, particulate matter, and scattering





coefficient. Furthermore, the underestimated rise of spectral albedo in 400–450 nm is caused by the rapid decrease on attenuation coefficient of the pure ice (Perovich et al., 1990; 1998). However, the increased proportion of pure ice in the

360   substrate ice will lead to a simulated result an order of magnitude higher than the observation, so an advanced understanding or method is required to solve the dilemma.

Despite that this model lacks the evaluation of spectral albedo from substrate ice, it is accurate and valid in the simulation of total albedo. Based on the observation dataset, the intervals of 0.01–0.1 m, 0.1–0.5 m and 1.1–1.8 m are taken as the distribution range for the ice lid thickness, melt pond depth and substrate ice thickness, assuming that it represents most melt

365   ponds from early August to early September. The basal value $A_0$ is 0.159 when all three variables are set to the minimum values. Then the ice lid thickness, melt pond depth and substrate ice thickness is gradually increased to the maximum respectively to obtain the variation of albedo with each property (Fig. 12a). It can be concluded that the albedo of ice ponds during freezeup is determined firstly by the ice lid, secondly by the substrate ice, and finally the pond depth. In addition, exponential fits for the total albedo and different properties are also provided, which may conduce to the albedo estimation

370   for melt ponds in certain state.

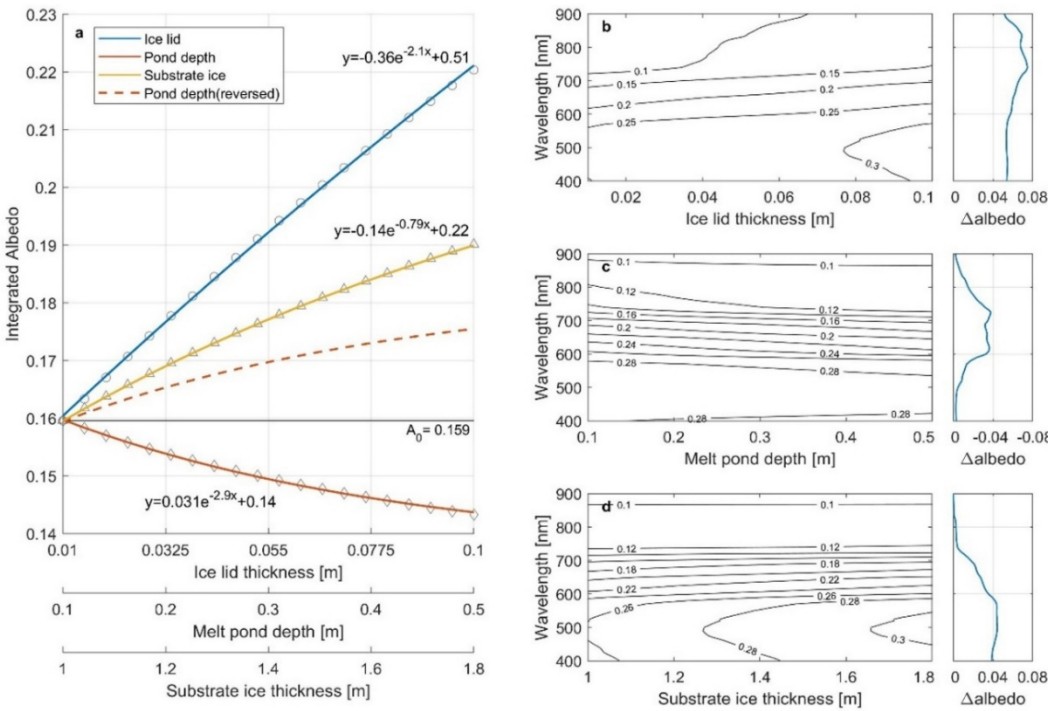

**Figure 12: Effects of ice lid thickness, pond depth and substrate ice thickness on (a) total albedo and (b)–(d) spectral albedo. In subplot (a), $A_0$ is the simulated value with the ice lid of 0.01 m, pond depth of 0.1 m and substrate ice of 1 m. The dots represent are the simulated albedo with the increasing of one variable only (circle for ice lid thickness, diamond for melt pond depth and**
375   **triangle for substrate ice thickness) and the lines are the fitting results. The representative range of the three variables are set consistent with the range of all observed values. The total variations of spectral albedo in subplots (b)–(d) is are plotted at the right side of subplot (b)–(d) them.**



Meanwhile, the physical properties regulate the spectral distribution of albedo in different ways (Figs. 12b–d). The increase of ice lid thickness leads to rise of albedo exceeding 0.05 at all bands, with nearly 0.08 between 700 and 900 nm, twice the other two properties. The influence of increasing pond depth on the reduction of albedo concentrates within the band of 600–800 nm. In contrast, the effect of substrate ice thickness is significant only in wavelengths less than 700 nm. The characteristics above may make it possible to infer more detailed changes in the physical properties of the melt ponds from the albedo.

## 4 Discussions

Compared to the melt ponds with open water surface, melt ponds with a frozen surface are less focused in former studies, while they make up over 90 percent of all the ponds in late August and early September in the Pacific Sector of the Arctic. During formation and evolution in June and July, the albedo of a single pond varies gradually with only one abrupt change at drainage (Light et al., 2022). Similar rate occurs in the early freezeup when the melt water freeze until the ice lid forms at the surface (Flocco et al., 2015), after which the albedo is strongly affected by the snow cover on it and therefore inferred to be highly variable, sometimes even higher than the unponded ice (Anhaus et al., 2021). This makes it difficult and less meanful to predict the albedo via simulation on thermodynamic processes only. Hence a statistics-based parameterization including more detailed fraction and typical albedo of different types at certain time is required to estimate the regional albedo during freezeup.

For these ponds with a closed surface, except for the influence of temperature and radiation which is discussed in section 3.2, causes such as precipitation and wind also have effects on the formation of them. Strong wind not only increases the turbulent heat flux on the surface of water pond and accelerate its shift to ice pond, but also brings snow to the top of ice lid, which is usually lower than the surrounding unponded ice (Anhaus et al., 2021; Light et al., 2022). Besides, rainfall and snowfall affect directly on the surface state of melt ponds (Niehaus et al., 2023), especially for those with ice lids or snow cover. However, the exact effects cannot be confirmed in this study due to the lack of continuous meterological data before observations and the absence of snow depth measurements, so is the quantitive analysis, which we will focus in the following studies. But still, the results on melt ponds with frozen surface in this study may be referentially valuable on the identification of snow-covered ponds in remote sensing and air-borne observation.

Following the same method in section 3.2, multiple combinations other than 360 nm and 490 nm were also examined and certain regularities are found for different ratios, for example, 405–420 nm and 662–672 nm which are band 8 and 13 of MODIS (hereinafter referred to as the central wavelength of the band, namely $\alpha_{412}$ and $\alpha_{667}$) in Fig. 13a and Fig. 13b. For albedo higher than 0.5, it can be similarly assumed that ponded sea ice shows a ratio of $\alpha_{412}/\alpha_{667}$ less than 1 and unponded sea ice shows the opposite. However, the ratio is larger than 1 for ponds with albedo lower than 0.5, so cautions should be taken when applying the ratio $\alpha_{412}/\alpha_{667}$ as an indicating status on whether the sea ice is ponded or not. Despite that, it can be used to differentiate snow pond and unponded ice, which is often challenging dealing with images. Furthermore, a linear




relationship is found in aspect of the mean value of $\alpha_{412}/\alpha_{667}$ that decreases with the freezeup of melt ponds by ~0.35 between adjacent types (Fig. 13b). And the ratio of unponded ice is 1.15 in the middle of ice-snow ponds and snow ponds. In conclusion, we show that it is possible to detect the surface states via spectral albedo, which is much required in the determination on the energy balance of the refreezing sea ice.

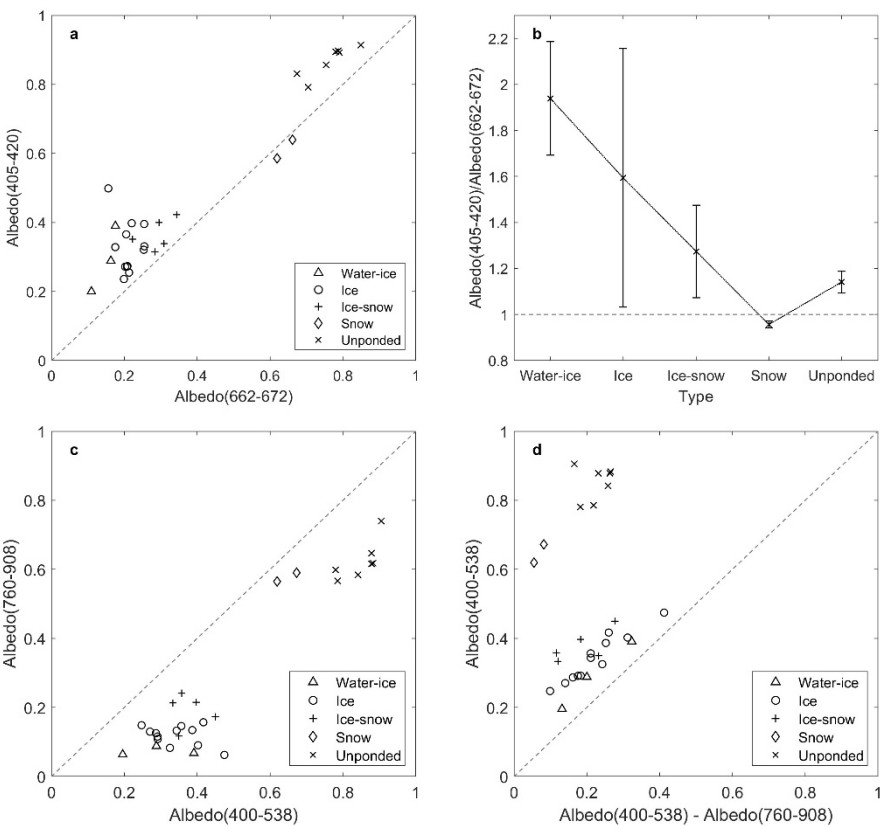

**Figure 13: The ratio of spectral albedo between 405–420 nm and 662–672 nm for (a) all ponds and (b) different types, and scatterplots for albedo ratio between different bands following (c) PCA and (d) LinearPolar Algorithm, where numbers in the brackets represent the limits of the integral albedo.**

The MPF (melt pond fraction) retrieval methods proposed by Rösel and Kaleschke (2011, named PCA Algorithm) and by Wang et al. (2020, named LinearPolar Algorithm) in which the band 2 (440–538 nm) and band 8 (760–908 nm) of Sentinel-2

are used is also examined as in Fig. 13c and 13d. Both algorithms fail to identify the snow ponds, which show a different spectrum compared to 'typical' melt ponds. There is a trend for clusters of different pond types that becomes gradually closer to that of unponded ice as melt ponds refreeze, yet a gap of 0.3 remains between them. However, the cluster of ice-snow ponds is treated as mixed pixel in the algorithms and associated with the transition of MPF between 0% and 100%, leading to an underestimation of real MPF during freezeup. Since snow pond is characterized by its relatively rapid increase

in the wavelength range of <550 nm and the slow reduction between 550 and 700 nm, band with shorter wavelength is required to ensure the distinction.

## 5 Summary

Based on the observed irradiance dataset and recorded physical properties of ponded and unponded sea ice during CHINARE 2012-2020, a detailed classification for refreezing melt ponds is proposed, and quantitative studies are made on
factors influencing the albedo of melt ponds.

The surface state dominates the albedo of refreezing melt ponds during early freezeup, and considerable distinctions are found between different types. To be specific, the typical albedo of water, water-ice, ice, ice-snow and snow ponds is 0.14, 0.20, 0.25, 0.39 and 0.74, respectively. As freezeup continues, factors such as air temperature and precipitation affect the surface of sea ice and change the surface condition of melt ponds. Therefore, melt ponds with snow cover (i.e. snow pond & 435 ice-snow pond) take up 10% in early August but exceeding 50% in early September, which contributes to the increase of average albedo from 0.27 to 0.36 with a rate of 0.0036 $d^{-1}$, slower than those observed in Amundsen Basin.

The spectral distributions of albedo of water-ice, ice and ice-snow ponds are similar, in contrast with that of snow-covered unponded ice. The albedo of snow pond is distinctive by its relative high values from other ponds and the increase at short wavelength (<550 nm) from unponded ice, enabling the effectivity of indicators such as $\alpha_{360}/\alpha_{490}$ and a$\alpha_{412}/\alpha_{667}$, which may 440 help to the further improvement of melt pond algorithms.

Under the same surface state, depth and ice thickness further determine the albedo. For instance, the growth of ice lid reduces the total albedo of ice ponds while increase that of ice-snow ponds, which possibly related to the formation of snow cover on the surface. Besides, parametrization of a radiative transfer model is improved based on the ice pond data and the remaining error from observation is therefore reduced. Further simulation shows that, within the observed range of
distribution, the importance of physical properties to total albedo is ranked as ice lid thickness, substrate ice thickness, and melt pond depth.

## Data availability

Data used in this study is included in the supplementary information.

## Author contributions

JZ had the concept of the study, processed and analyzed the data, and wrote the draft of manuscript. TL contributed to the concept of the study and participated in discussions of the result. PL provided the radiative transfer model of melt pond. TL,



YL and XW deployed the instruments in the field. All authors provided critical feedback and contributed to the editing of the manuscript.

**Competing interests**

The authors declare that they have no conflict of interest.

**Acknowledgements**

Melt pond data used in this manuscript was produced during the Chinese National Arctic Research Expeditions. We are deeply grateful to the organizers, crews and scientists onboard the icebreaker R/V Xuelong and R/V Xuelong 2 during the Chinese National Arctic Research Expeditions. We acknowledge the reviewers and the editor for their comments and 460 suggestions.

**Financial support**

This study was supported by the National Key Research and Development Program of China (2021YFC2801103) and the National Natural Science Foundation of China (42276239).

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
