# Peer review of "The dependence of albedo on different factors for refreezing melt ponds in the Arctic"

_EGUsphere, 2025_

## Author Response (AR1)

**Response to the comments of Editor and Reviewers**

Please find our response to the comments in **blue** font under the respective comments.

We thank the editor and the reviewers for the time and effort to thoroughly read and evaluate our manuscript, and we are grateful to the editor and the reviewers for the constructive comments and criticism which help us to improve our work. We have taken all points into account and revised the manuscript accordingly. In the following we give point-by-point responses to all the comments, and a list of the relevant changes in the end of this file.

The places of changes both in the revised version and mark-up version are attached for your information.

**Editor**

Received: 28 October 2025

To what extent could impurities in the snow have affected the spectral albedo, and could this undermine you conclusion that it is possible to detect the surface states from spectral albedo?

Thank you for your comments which points out the effect of impurity in the snow. The major light-absorbing snow impurities in the Arctic include soot (black carbon, BC, or elemental carbon, EC), organic carbon and soil dust (Warren, 1982; Bond et al., 2013). The equivalent EC, i.e. the amount of EC that would need to be present in the snow in order to account for the observed absorption, is defined and commonly used to describe the light-absorbing impurities in snow. The median of equivalent EC of snow on sea ice in the Arctic Ocean in summer is 14±15 ng $g^{-1}$ as reported in Doherty et al. (2010), and the impurity at a concentration of 30 ng $g^{-1}$ causes a reduction of 0.8% on integrated albedo (Pedersen et al., 2015).

According to Warren (2013) and Hadley et al. (2012), the spectral albedo of pure snow decreases as the wavelength increases in 300–800 nm, while impurity in snow reduces the albedo in ultra-violet band, causing the appearance of albedo maximum and its shift towards 600 nm at high concentration (>100 ng $g^{-1}$), as shown in Figure S1b and Figure S2a below. Due to the low concentration (14±15 ng $g^{-1}$) in the Arctic Ocean (Doherty et al., 2010), the spectral albedo of snow shows minor variation between 300 and 600 nm compared to pure snow, and the pattern that albedo decreases with

increasing wavelength still exists. This is consistent with the albedo characteristics of Arctic snow-covered sea ice observed in this study, as well as those obtained from observations and simulations in previous studies (Warren and Wiscombe, 1980; Malinka et al., 2016; Light et al., 1998, 2022), as shown in Figure S3–S6 below. In this study, we take $\alpha_{360}/\alpha_{490}<1$ as the indicator of melt pond so our conclusion is reliable for typical snow with low impurity concentration in the Arctic Ocean. The appearance of anomalous dirty snow with impurity over 100 ng g$^{-1}$ would affect the validity of the conclusion, but snow with such high concentration was mostly observed near the cities and has barely been observed in the Arctic Ocean (Doherty et al., 2010; Pedersen et al., 2015).

In some occasion, there might be exceptional case such as wind crust (Figure S6a below) or large grain (Figure S2b and S2c below) of snow cover, which cause minor albedo peak around 500 nm and disable the ratio $\alpha_{360}/\alpha_{490}$. But since the albedo of these cases (~0.9 in 400–600 nm) is much higher than that of snow pond (<0.7), it can be identified simply by the value of albedo. Moreover, those exceptional cases occur mostly in late autumn, when the air temperature is low enough to maintain the existence of dry snow. Therefore, our conclusion that it is possible to detect the surface states from spectral albedo is tenable for Arctic sea ice in late summer and early autumn. Similarly, the snow thickness on the surface of snow pond and the water trapped in the pond both fall within a certain range for most cases, where our conclusion is reliable. But if we want to expand it to other regions or a longer period, the uncertainties need to be further investigated, and this is also the focus of our subsequent study in the future.

We added some detailed discussion about the uncertainty of our result in Section 3.2 as follows, and emphasized the importance of uncertainties including snow impurity in Section 5.

"The continuous reduce of albedo with the increase of wavelength indicates low concentration of impurities in the snow cover (Warren, 2013), which is also the typical characteristic of Arctic sea ice in early autumn (Light et al., 2022)." **(Line 280–282)** (Line 310–312 in mark-up version)

"Besides, some uncertainties remain such as snow impurity, snow depth and pond water, which affect the spectral distribution of albedo for snow pond and unponded ice but have minor effect on this result. For instance, soot in snow reduces albedo in ultra-violet but do not cause albedo increasing along with the wavelength in 400–600 nm at a low concentration (<100 ng g$^{-1}$ while it is 14±15 ng g$^{-1}$ in Arctic Ocean) of elemental carbon (Doherty et al., 2010; Warren, 2013). Anhaus et al. (2021) reported refrozen pond with thicker snow cover and consequently higher albedo than adjacent ice, but

the rare situation requires continuous strong wind exceeding the threshold of snow drifting (8–10 m s$^{-1}$) for weeks. In addition, pond water largely contributes to the distinctive characteristics of snow pond so the pond becomes harder to identify as pond water refreezes. The result can be expanded and applied to an extended period by quantifying these factors, and it may help on the further development of melt pond detection algorithm." **(Line 311–319)** (Line 343–351 in mark-up version)

"The melt pond dataset collected from five Arctic expeditions was used in this study, but we were still limited by the incomplete measurement in some stations and the short of sample numbers for several types of ponds, especially the snow pond. Hence the uncertainties of pond albedo (i.e. weather condition, snow impurity, snow thickness and pond water) remain unclear, requiring detailed records on meteorological, chemical and physical properties to enhance current result." **(Line 484–487)** (Line 520–523 in mark-up version)

And we added the cited paper into the part "Reference" of the revised manuscript.

[Figure]

**Figure 1.** Comparison of the spectral signature of snow thinness to that of black carbon (BC) in snow, for snow grain radius $r_e = 1$ mm and solar zenith angle $\theta_0 = 60°$. (a) Spectral albedo of pure snow over a black surface for a variety of snow depths expressed in liquid equivalent. The top curve is for semi-infinite depth. Redrawn from Figure 13c of *Wiscombe and Warren* [1980], using updated optical constants of ice [*Warren and Brandt*, 2008]. (b) Spectral albedo of deep snow containing various mixing ratios of BC in parts per billion by mass (nanograms of BC per gram of snow). Redrawn from Figure 7b of *Warren and Wiscombe* [1980], using updated optical constants of ice and BC. The optical constants and size distribution for BC used in the model are those described by *Brandt et al.* [2011]. The digital values for this figure are available at http://www.atmos.washington.edu/ice_optical_constants.

**Figure S1. The effect of snow thickness and black carbon on the albedo of snow, in Warren (2013).**

[Figure]

**Figure 1 | Spectral albedo of snow of different $R_{eff}$ measured in our experiments (dots) and modelled using SNICAR (shaded bands). a,** 55 μm $R_{eff}$, **b,** 65 μm $R_{eff}$ and **c,** 110 μm $R_{eff}$. Error bars show the standard deviation of the measurements. Upper and lower boundaries of the shaded bands correspond to modelled albedo assuming BC mass absorption cross-sections, at $\lambda = 550$ nm, of 7.5 and 15 m$^2$ g$^{-1}$, respectively.

**Figure S2. Spectral albedo of snow with different $R_{eff}$ (optical effective radii) and black carbon, in Hadley (2012).**

[Figure]

FIG. 8. Combined effect of finite depth and soot content on snow albedo $a_d$.

**Figure S3. The snow albedo with soot content, in Warren and Wiscombe (1980).**

[Figure]

**Figure 2. Representative spectral albedos for individual ice surface types.** Measurements include $\alpha_\lambda$ curves and range of $\alpha_{tot}$ values (inset legend) for eight individual surface types. Thin ice and lidded pond values represent example values; full range of values not specified. Photographs show measurement sites corresponding to each curve

**Figure S4. Spectral albedo of different surface types observed during MOSAiC, in Light et al. (2022).**

[Figure]

**Figure 9.** Typical white ice in Arctic summer. The surface is slightly wet; the grains are larger than those in dry white ice; the scattering layer is 12–18 cm thick. The average air temperature is $-1.1\,°C$.

**Figure S5. The spectral albedo of sea ice with thin snow, in Malinka (2016).**

[Figure]

**Figure 12.** Three special cases: thin wind crust on top of fine fresh snow of 4 cm thickness (left panels), a frozen-over gray melt pond with snow on top (middle panels), and a frozen-over crack with air bubbles and algae inclusions (right panels). The average air temperature is $+0.3\,°C$ (right panels) and $-1.6\,°C$ (the others).

**Figure S6. The spectral albedo of different surface of sea ice, in Malinka (2016).**

**Reference**

Anhaus, P., Katlein, C., Nicolaus, M., Hoppmann, M., and Haas, C.: From bright windows to dark spots: Snow cover controls melt pond optical properties during refreezing, Geophys. Res. Lett., 48, e2021GL095369, doi:10.1029/2021GL095369, 2021.

Bond, T. C., Doherty, S. J., Fahey, D. W., Forster, P. M., Berntsen, T., DeAngelo, B. J., Flanner, M. G., Ghan, S., Kärcher, B., Koch, D., Kinne, S., Kondo, Y., Quinn, P. K., Sarofim, M. C., Schultz, M. G., Schulz, M., Venkataraman, C., Zhang, H., Zhang, S., Bellouin, N., Guttikunda, S. K., Hopke, P. K., Jacobson, M. Z., Kaiser, J. W., Klimont, Z., Lohmann, U., Schwarz, J. P., Shindell, D., Storelvmo, T., Warren, S. G., Zender, C. S.: Bounding the role of black carbon in the climate system: A scientific assessment, J. Geophys. Res. Atmos., 118, 5380–5552, doi:10.1002/jgrd.50171, 2013.

Doherty, S. J., Warren S. G., Grenfell T. C., Clarke A. D. and Brandt R. E.: Light-absorbing impurities in Arctic snow, Atmos. Chem. Phys., 10(23), 11647–11680, doi:10.5194/acp-10-11647-2010, 2010.

Hadley, O. and Kirchstetter, W.: Black-carbon reduction of snow albedo, Nat. Clim. Change, 2, 437–440, doi:10.1038/nclimate1433, 2012.

Light, B., Eicken, H., Maykut, G. A., and Grenfell, T. C.: The effect of included participates on the spectral albedo of sea ice, J. Geophys. Res. Oceans, 103, 27739–27752, doi:10.1029/98JC02587, 1998.

Malinka, A., Zege, E., Heygster, G., and Istomina, L.: Reflective properties of white sea ice

and snow, The Cryosphere, 10, 2541–2557, doi:10.5194/tc-10-2541-2016, 2016.

Pedersen, C. A., Gallet J.-C., Ström J., Gerland S., Hudson S. R., Forsström S., Isaksson E. and Berntsen T. K.: In situ observations of black carbon in snow and the corresponding spectral surface albedo reduction, J. Geophys. Res. Atmos., 120, 1476–1489, doi:10.1002/2014JD022407, 2015.

Warren, S. G. and Wiscombe, W.: A model of spectral albedo of snow, II: Snow containing atmospheric aerosols, J. Atmos. Sci., 37, 2734 – 2745, doi:10.1175/1520-0469(1980)037<2734:AMFTSA>2.0.CO;2, 1980.

Warren, S. G.: Optical-properties of snow, Rev. Geophys., 20(1), 67–89, doi: 10.1029/RG020i001p00067, 1982.

Warren, S. G.: Can be black carbon in snow be detected by remote sensing?, J. Geophys. Res., 118, 779–786, doi:10.1029/2012JD018476, 2013.

**Anonymous Referee #1**

**Abstract:**

The statement of the research objective should be clearly articulated. The goal of investigating the optical and physical properties of refreezing ponds must be explicitly stated. What exactly is the spectral range? What are the surface characteristics of the ice-cover? This is not entirely clear in its current form. The abstract should clearly communicate that the primary objective is to generate a dataset focused on albedo-based classification of surface states, with particular emphasis on refrozen melt ponds (which remain under-documented compared to seasonal albedo variations). The exact goal should be distinctly stated—**what** and **why**. If there are multiple objectives, they could be presented in order of priority. For example, the development of the methodology could be considered one of the objectives, along with the spatial and temporal variation of albedo and the relationship between physical and optical properties.

Thanks for your constructive comments. We have rephrased the abstract to clarify the goals of this study and to add more specific information of the observation. The revised abstract is attached below: **(Line 9–19)** (Line 20–28 in mark-up version)

"Sea ice plays an important role in the heat transfer into the Arctic Ocean whereas the presence of melt ponds on sea ice complicates the scenario. However, the refreezing pond is less focused and documented in comparison with the well-established seasonal variation. To better evaluate the effect of melt pond on the freezeup of sea ice, we conducted a series of observations with 81 melt ponds in the central Arctic during fall freezeup, 2012–2020. The melt ponds are categorized into five types based on the surface state to effectively investigate the various characteristics. The total albedo of each type is 0.14 (water pond), 0.20 (water-ice pond), 0.25 (ice pond), 0.39 (ice-snow pond), 0.74 (snow pond), respectively, showing the increase on albedo in August and September (0.0036 $d^{-1}$) due to the changes of the surface state. The albedo dependence on the surface state, ice lid, pond depth and underlying ice is examined using both in-situ measurements and modified radiative transfer model, with result indicating the dominance of surface state followed by ice lid thickness. The total albedo of ice ponds decreases with increasing pond depth, and the raising of ice lid thickness reduces the albedo while rises that of ice-snow ponds. In addition, further analysis reveals the capacity of different ratios of spectral albedo on the distinction between snow-covered pond and unponded ice, potentially improving the melt pond retrieval algorithms."

**General comments:** The albedo categorization is unclear. What is the basis for the threshold? Since the distinction of 'classes' or types is vital, it may benefit from a brief explanation of the rationale behind this threshold. Why does refreezing specifically matter, and how does your classification fill a particular gap?

In this study, melt ponds are classified based on their surface states. We apologize for the misleading sentence in the abstract (Line 12, rewritten in the revised version), which was tend to emphasize the albedo difference between various types of melt pond, but not to describe the method of categorization. The surface state of each melt pond was classified based on both the description in the station documents and the recorded images during observation. To be specific, the state of melt ponds within a radius of 1–2 meters (depends on the field of view of sensors) centered on the projected position of the optical sensor is taken into consideration.

Previous studies have conducted extensive observations and analyses on the formation and evolution of melt ponds from June to July, and the temporal variation of melt pond albedo during this period is well understood. However, the surface states of melt ponds are almost always water during this stage. As a result, current understanding of ice lid and especially snow cover on melt ponds, which is less focused, is relatively limited, causing a lack of capacity to quantify the temporal variation of melt pond albedo during refreezing. However, melt pond is one of the main factors on the albedo variation of sea ice in this period, since unponded ice has similar albedo due to the snow cover later than mid-August. To gain more comprehensive understanding of sea ice, study on melt pond during refreezing is necessary.

Considering that albedo of individual melt pond is affected by weather such as wind and snowfall, thus it is hard to effectively predict its temporal variation during refreezing. But classifying the refreezing ponds allows us to acquire the range and the typical albedo of each type. By combining the proportions and albedo of different types, the temporal variation of the overall albedo of melt ponds can be better evaluated. Therefore, the study will enhance the understanding of sea ice albedo in the Arctic, and will contribute to the simulation or prediction on temporal variation of sea ice.

Based on the content above, we added more explanation about the categorization in Section 2.3 **(Line 146–147)** (Line 170–171 in mark-up version), and we added sentences in Section 5 **(Line 487–491)** (Line 523–527 in mark-up version) about the significance and potential use of this study.

**Introduction:**

- **Line 35:** "The surface albedo of sea ice in summer is largely determined by the melt ponds" — This is an overstatement, as bare ice areas still contribute significantly.

- Thanks for the correction. This sentence has been changed to "...the surface albedo of sea ice in summer is significantly affected by the melt ponds". **(Line 37)** (Line 46–47 in mark-up version)

- **After Line 45:** I would suggest adding 1-2 sentences on why the refreezing period specifically matters (e.g., the transition from net melting to ice growth and model parameterization challenges during shoulder seasons). This addition would provide better context and emphasize the importance of documenting this stage. This also connects more smoothly to the section discussing existing knowledge gaps.

- We have added some sentences to explain the importance of refreezing period at the end of Line 45: "Thus, the thermal characteristics of summer sea ice strongly depends on fraction of melt ponds, and it remains remarkable during refreezing period when the transition from ice melting to ice growth occurs. The ice growth at the base of the underlying ice is limited due to latent heat stored in pond water which is trapped by ice lid until the pond freeze completely (Flocco et al., 2015). Furthermore, these various thermal processes also pose challenges to the accurate simulation of models." **(Line 47–51)** (Line 57–60 in mark-up version)

**In-situ Data:**

- The comprehensive spatial coverage, especially from multiple Arctic basins, is commendable.

- Thank you for your recognition. We will further expand the coverage of this dataset in subsequent observations and studies.

- **Line 66-67:** Could you briefly explain the reasoning behind the constraint imposed on data collection (overcast skies), defined by visual observation? Were clear-sky conditions excluded? If so, could you explain why, since this would exclude direct-beam scenarios?

- Yes, the data obtained under clear-sky are excluded and there are several reasons. Firstly, both some of the authors and the former reviewers suggest that the albedo of melt ponds under diffusive light and direct light differs from each other. Therefore, the data used in this study should be measured under the same light condition. Secondly, most ice stations were set during overcast weather since it's overcast at most time of the mentioned Arctic expeditions in this study. The melt ponds observed under clear-sky are too few (0~2 each

cruise, 5 in total) to get convincing results. In order to well illustrate the characteristic of melt ponds, observations under diffusive light were used. Besides, the time with clear skies is only 6% in August and September in the Arctic (Grenfell and Perovich, 2008), so the albedo under diffusive light is more representative of melt ponds in this period.

- **Line 35:** The term "Surface scattering layer" — It might help to provide a brief definition for non-specialist readers.

- A brief definition has been added to help understanding this term: "a thin upper layer of sea ice with high scattering capacity for incoming solar radiation". **(Line 36)** (Line 45–46 in mark-up version)

**Methodology:**

- **Lines 65-66:** Was the constraint on data collection (overcast skies) based on visual observation? If clear-sky conditions were excluded, could you explain why, since this would exclude direct-beam scenarios?

- The observations were conducted no matter the weather was sunny or overcast. But the only data under overcast conditions were selected (based on the in-situ document, both visual record in pictures and weather record in texts) in this study to avoid the distraction on albedo caused by the difference between diffusive light and direct light. Please also see the detailed reply to comment "Line 66-67" in part "In-situ Data"

- **Lines 84-87:** It should be clearly stated which physical properties were measured (and how), and which instruments were used to measure ice lid thickness and pond depth. This would improve reproducibility and help clarify the relevance and quality of the collected data.

- Thanks for your comment. According to the in-situ record, ice lid thickness was measured by a metal ruler, the pond depth was measured by a metal tape measure and the substrate ice thickness was measured by an ice thickness gauge. The detail of measurement on physical properties such as ice lid thick or ponds depth has been added: "the ice lid thickness was measured by a metal ruler, the pond depth was measured by a metal tape measure, and the substrate ice thickness was measured by an ice thickness gauge". **(Line 92–94)** (Line 101–104 in mark-up version)

- **Line 63:** Multiyear ice is mentioned. Were all stations clustered in certain

regions or ice types? Could this potentially impact the representativeness of the data?

- The locations of ice stations were randomly picked and distributed in different regions (see Figure 1) so there is no impact on the representativeness of data at this aspect. All stations were set on multi-year ice, since seasonal ice in August or September is too thin to allow ice field operations on it. And the refreezing process of melt pond, especially the variation of surface condition is slightly affected by the age of sea ice since it depends mainly on the weather condition and events, so observations on multi-year ice still provide data reliable enough to support the results in this study.

- **Line 86:** Were locations randomly selected? If there was a deliberate choice behind location selection, could this introduce bias or affect the representativeness of the data?

- All the locations of ice stations were randomly selected with no deliberate choice.

- **Table 1:** It would be helpful to include information on which year each instrument was used (e.g., CNr4 measurements applied in 2012, 2014, and 2016). In the text, this could be elaborated to clarify the distinction between different radiometers and their corresponding wavelength ranges.

- We added information of instrument and the corresponding wavelength in the caption of Table 1 to clarify. **(see the caption of Table 1)**

- **Lines 164-65:** The calibration process is improved by using different coefficients for different surfaces. However, is there a potential circular dependency? That is, could there be a risk of misclassifying a pond initially, which would then lead to the application of the wrong calibration coefficient, reinforcing the misclassification? Also, how confident is the visual classification? Could this be slightly subjective? It might be worth exploring this further.

- Thank you for the comments. As mentioned before, the classification is based on the description in the station documents and the recorded images during observation, not the value of albedo. That is, the change on albedo does not affect the classification, so there is no potential circular dependency. The description in the station documents recorded by field operators may be slightly subjective but we also double-checked and classified all ponds based on photos recorded during observation (which will be provided as part of the dataset) as objective evidence.

- **Lines 165:** Water ponds or water-ice ponds? It might be useful to clarify this distinction.

- The calibration in Section 2.3 is aimed to solve the problem of poor consistency between the two instruments in the latter three categories (i.e. ice pond, ice-snow pond and snow pond). It can be seen that the deviation of the two instruments increases along with the formation of ice and snow, so the deviation of water-ice pond (<0.01) is even smaller than the average deviation of the three calibrated types (ice, ice-snow and snow) after calibration (0.049, 0.017, 0.104). In this case, we did not perform calibration for water-ice pond. Besides, the water ponds were only observed by one instrument so no calibration is performed.

**Results:**

- **Lines 176-177:** It would be preferable to avoid speculating about the reduction or slowdown unless statistical analysis is used to account for differences.

- Thank you for pointing this out. We removed the speculation about the annual reduction and only retained the objective comparison of different observations. **(Line 208–211)** (Line 237–240 in mark-up version)

- **Figure 6:** Consider increasing the font size at the top of the graph for better readability.

- We increased the font size of Figure 6a.

- **Lines 189-194:** Could you clarify whether these percentages are based on the number of observations or area coverage?

- The percentages referred here are based on the number of observations. We added a statement to make it clear for readers. **(Line 201)** (Line 230 in mark-up version)

- **Section 3.1:** A clearer separation between observations and interpretations would be beneficial. Consider distinguishing between: 1) Measured albedo changes; 2) Observed surface state changes and 3) Mechanism hypotheses

- Thanks for your valuable suggestion. We reorganized part of the sentences accordingly in Section 3.1 to improve the readability. **(Line 190, 208–211)** (Line

214–219, 237–240 in mark-up version)

- **Line 215-217:** Why emphasize the SHEBA comparison if ice type is a key variable? It seems that since 0.14 is equidistant from 0.12 and 0.2 but considerably different from 0.4, it might be better to state that the results align more closely with the dark pond literature, rather than general melt pond values.

- We are sorry for the confusion caused by a missed key description here. The value 0.4 refers to the albedo of light ponds observed in SHEBA, while the albedo of dark ponds observed at the same ice station is ~0.2, which is consistent with other observations mentioned later. We corrected the statement to emphasize the alignment with dark pond literature. Besides, the result in SHEBA also shows that sea ice type is not the dominant factor in the albedo of melt ponds. The mention of sea ice type here is intended to describe the observation detailly, but now we have realized that it is unnecessary and may cause ambiguity, so we removed description about ice types in this statement. **(Line 231–233)** (Line 260–263 in mark-up version)

- **Line 237-238:** Please carefully identify the source of noise. It is important to accurately describe instrumental limitations rather than referring to solar physics.

- This sentence is intended to state that the irradiance in ultra-violet wavelength is close to the minimum resolution of the instruments, thus the instrumental limitation caused the noise. We rewrite the statement to better describe this: "Since less energy of solar radiation concentrates in ultra-violet wavelength, approaching the minimum resolution of instruments and therefore leading to noises in 320–350 nm...." **(Line 253)** (Line 283–284 in mark-up version)

- **Lines 385-393:** You claim that >90% of ponds are frozen by late August/early September, yet Figure 6a shows only around 30% ice ponds and approximately 50% ice-snow/snow ponds by early September (Lines 190-191). Please reconcile these numbers or clarify what "frozen surface" refers to—are water-ice ponds counted as frozen?

- We apologize for the misunderstanding caused by the ambiguous meaning of the term "frozen" here. The use of "frozen" is intended to distinguish ponds with surface partly or completely covered by ice lid (both bare and snow-covered) from ponds with completely unfrozen surfaces, which have been extensively studied. Therefore, water-ice ponds are also counted as frozen. We changed the description to "ponds with surface partially or completely frozen" to clarify. **(Line 416)** (Line 448 in mark-up version)

- **Lines 418-424:** The critical finding that existing MPF algorithms (PCA, LinearPolar) fail to identify snow ponds and underestimate MPF during freeze-up is currently buried. This finding could be elevated as a main result rather than being relegated to the discussion section.

- We thank you very much for your positive evaluation. However, only two commonly-used melt pond retrieval algorithms are compared so the evaluation is not comprehensive for all current algorithms. Additionally, the number of snow pond observed is kind of small to provide a more robust result, which is also a limitation we regret. For the reasons mentioned above, the statements here is not sufficient enough to become part of the results as we evaluated, so the purpose of this section is to explore the misdetection issue of commonly used algorithms on snow ponds and provide potential directions of improvement. Thus, it is more appropriate to be included as part of the discussion section. We will further investigate the limitation and the potential improvement on MPF algorithms in our future work.

- Besides, to draw readers' attention to this content, we also mentioned and emphasized it in the conclusion section as follows: "The spectral distributions of snow pond is distinctive by its relative high values from other ponds and the increase at short wavelength (<550 nm) from unponded ice, causing misidentifications of MPF algorithms but also enabling the effectivity of indicators such as $\alpha360/\alpha490$ and $\alpha412/\alpha667$, which may help to the further improvement of melt pond algorithms". **(Line 474–477)** (Line 510–513 in mark-up version)

- **Lines 441-445:** Could the error from observations (which your model improves upon) be quantitatively shown?

- We added quantitative results about error from observations in Figure 11a. The statement about reduced error is also added in Section 3.4 and in the summary: "…the average RMSE (root mean square error) of all ponds reduces from 0.168 to 0.026 after the modification". **(Line 481)** (Line 517 in mark-up version)

- **Lines 429-440 (Summary):** The summary appears to introduce new information that was not present in the results section. For instance, the comparison of slower observations in the Amundsen Basin (Line 437) seems to be new. A summary should synthesize previously presented findings.

- Thanks for pointing out this issue. We added statement about comparison with

observations in Amundsen Basin in Section 3.1. **(Line 208–211)** (Line 237–240 in mark-up version) Besides, we also checked the rest of the summary section to ensure every statement is introduced in result section before presented here.

- **Line 439:** Typo: "aα412/α667" should be corrected.

- Thank you for pointing this out. We have corrected accordingly. **(Line 476)** (Line 513 in mark-up version)

**Anonymous Referee #2**

**General comments**

The manuscript from Zhu et al. identifies the role of ice-lead thickness, melt-pond depth, and substrate ice thickness in total and spectral albedo values of melt ponds. Melt ponds have a large impact on sea ice albedo during the summer and fall. Moreover, their characteristics also differ during the refreezing season (August-September) compared to the fast-melting period (June-July). Therefore, this study addresses relevant aspects of Arctic sea ice life cycle. Results from the observations of total and spectral albedo, along with the radiative transfer simulations, are well described with comprehensive step-by-step explanations. However, some effort is required to introduce the radiative transfer model, to avoid copying and pasting equations and text from Lu et al. (2016), and instead to highlight what was adapted in the model for this specific analysis. The discussion section will also benefit from a comparison between the observations and satellite measurements (see specific comments). The summary should be expanded to a conclusion that identifies the strengths and limitations of this study and outlines the next steps to advance the analysis.

Thanks for your positive evaluations and your constructive comments, which help us to further improve this work. We rewrote Section 2.2 to avoid meaningless repetition and to emphasize the modification we made to the model, added explanation on the underlying principles of melt pond algorithms and the potential direction of improvement and expanded the summary to a conclusion with highlights, limitations and prospects of this study. For more detailed answers to each point, please see our responses under specific comments.

**Specific comments**

- **Line 21:** "and thus control the radiative forcing in the Arctic Ocean and the world (Hudson, 2011)". Would it be possible to rephrase the sentence for more accuracy, avoiding "the world" which sounds simplistic in the context of the paper?
- We rewrote the sentence to improve the accuracy: "thereby regulating the radiative forcing within the Arctic Ocean and throughout the global climate system". **(Line 22)** (Line 32 in mark-up version)

- **Line 65:** "short-term and long-term ice stations" could these stations be identified in Table 1 or in Figure 1? Did you notice significant discrepancies in total albedo or spectral albedo due to the sampling duration?

- Whether an ice station is short-term or long-term can be identified based on the third column "Date" of Table 1, where the long-term ice station spanning multiple dates (only IC2004 in this study). Besides, since all observation data used in this study were collected under overcast skies with diffusive light, the impact of solar elevation angle on albedo is negligible. Therefore, sampling duration does not significantly affect total albedo or spectral albedo.

- **Line 92:** lambda in the equations should be defined.
- Corrected accordingly. **(Line 116)** (Line 132 in mark-up version)

- **Section 2.2** this section has to be significantly reworked to specify what differs from than Lu et al. (2016) and to clearly identify the novel elements of the present analysis.
- Thank you for your comments. We rewrote Section 2.2 to avoid simple repetition of Lu et al. (2016) and to emphasize the modifications we made to the model. The revised version starts with the irradiance of each layer to introduce and highlight the role of different coefficients in the model, after which the modifications we adopted to the parameterization of origin model are introduced and the reasons and references of these modifications are explained. These changes to the parameterization of the model better represent the optical properties of the Arctic sea ice in the refreezing period, thus effectively improving the accuracy of the simulation results.
- The rewritten part is attached below: **(Line 113–142)** (Line 125–166 in mark-up version)

"In this model, sea ice is treated as isotropic under the assumption of diffuse incident solar irradiance. The upward and downward irradiance of each layer can be described as in (Lu et al., 2016):

$$\begin{cases} F^{\downarrow}(z,\lambda) = A(1 - \mu_\lambda)\exp{(\kappa_\lambda z)} + B(1 + \mu_\lambda)\exp{(-\kappa_\lambda z)} \\ F^{\uparrow}(z,\lambda) = A(1 + \mu_\lambda)\exp{(\kappa_\lambda z)} + B(1 - \mu_\lambda)\exp{(-\kappa_\lambda z)} \end{cases}, \tag{3}$$

where $z$ is depth in certain layer, $\lambda$ is wavelength, $F^{\downarrow}(z, \lambda)$ represents downward irradiance, $F^{\uparrow}(z, \lambda)$ represents upward irradiance, $A$ and $B$ are constants determined by the boundary conditions, $\mu_\lambda$ represents the absorption strength (0 for purely scattering medium and 1 for purely absorbing medium), and $\kappa_\lambda$ represents the attenuation coefficient. As defined in Perovich (1990), and can be written as

$$\mu_\lambda = \sqrt{k_\lambda/(k_\lambda + 2\sigma_\lambda)}\ , \tag{4}$$

$$\kappa_\lambda = \sqrt{k_\lambda(k_\lambda + 2\sigma_\lambda)}\ , \tag{5}$$

where $k_\lambda$ represents absorption coefficient dependent on wavelength and $\sigma_\lambda$ represents the scattering coefficient as a constant independent of wavelength. The Fresnel reflection coefficient between water and ice is neglected and the reflection at the air-water interface is taken as 0.05 for the diffuse sky, according to Perovich et al. (1990).

In this study, we adopted several modifications to the origin model. Firstly, the band of incident solar irradiance $F_0(\lambda)$ is set to 400–900 nm based on the range of in-situ measurements and the band of coefficients reported in previous studies. Secondly, the parameters of the inherent optical properties including absorption and scattering coefficients for the substrate ice are modified based on the field record to ensure the simulation to be consistent with the observation. Wang et al. (2020b) reports that the volume of bubbles and brine varies oppositely with the increasing of depth, causing inhomogeneous optical properties of the ice beneath melt pond. Here a combination of attenuation coefficient for white ice interior and pure ice in Perovich et al. (1990) is adopted, instead of that for pure ice used in original settings. According to Perovich et al. (1990), the scattering coefficient of white ice interior is 2.5 m$^{-1}$, while Light et al. (2015) argue that the scattering coefficient of substrate ice varies between 10 and 22 m$^{-1}$, and a value of 13 is taken in the multi-layer model (Light et al., 2008). In this study, as most of the melt ponds observed are dark ponds and the resulted high scattering coefficient is one order of magnitude higher than the observed, so the scattering coefficient of substrate ice is set to 2 m$^{-1}$, consistent with Malinka et al (2018) and Katlein et al. (2015). Besides, the incident irradiance, ice lid thickness, pond depth and substrate ice thickness are all adopted from the in-situ observation."

- **Figure 3:** it should be acknowledged in the figure legend that it has been adapted from Lu et al. (2016).
- Corrected accordingly. **(see caption of Figure 3)**

- **Lines 112:** the equations are exactly the same than in Lu et al. (2016) along with the description until line 117. It should be acknowledged. It should also be better to highlight how adding the ice lid is impacting the equations and what is different in the calculations compared to previous studies.
- We acknowledged the equations and description from Lu et al. (2016) and added statements about our modifications adapted to the original model. **(Line 114)** (Line 126 in mark-up version)

- **Line 119:** in which equations/calculation R1 and R2 are used in the analysis?
- Both R1 and R2 are parameters within the model's parameterization. In this study, they were only used in the calculation process of the modified model, and were

not applied in other equations. We rephrased the sentences to clarify the features of the model and avoid confusion they may cause. **(Line 124)** (Line 145 in mark-up version)

- **Line 167:** "As a result, the calibration reduces the median deviation of CNR4 measurements from 0.2 to 0.06" is this reduction applied to all types of melt ponds? Maybe add some values in Fig. 5c and 5d to highlight the impact of the calibration.
- The reduction in deviation mentioned in this sentence is the average value obtained from the three types of melt ponds (ice pond, ice-snow pond, snow pond) that underwent calibration. We added the mean values (not median value since there are only several samples for some type) of deviation between two instruments for different types before and after calibration in Figure 5 to show the impact of the calibration. **(see Figure 5)**

- **Figure 6b:** what does the colorbar represent? What about the numbers and uncertainty/standard deviation in red? MP96, IA94, etc. are not defined yet.
- Thank you for the comment. We apologize for the missing of explanations for annotations in Figure 6b. The colorbar in Figure 6b represents the albedo and the numbers in red represents the average value with standard deviation in certain region. Acronym such as MP96 and IA94 represents albedo in previous observations. These explanations have been added in the caption of Figure 6: "In panel (b), the colorbar represents the albedo of melt pond, the annotations in red represent mean and deviation of pond albedo in certain regions, the annotations in black represent mean and deviation of pond albedo reported in previous studies, where the acronym is as follows: GM77 – Grenfell & Maykut, 1977; IA94 – Ivanov & Alexadrov, 1994; ML96 – Morassutti & Ledrew, 1996; MP96 – Makshtas & Podgorny, 1996". **(see the caption of Figure 6)**

- **Line 278:** "which is consistent with observation in Malinka et al. (2016)" Can you elaborate more on the agreement between your observations and previous studies?
- There was a snow-covered melt pond (the middle panel of Figure S7 below) observed during PS80/335 as reported in Malinka et al. (2016), with its spectral albedo between 730 and 950 nm showing a similar pattern with that of snow ponds observed in this study (Figure 8a in the manuscript). In comparison, spectral albedo of the two frozen ponds without snow (panel (a) and (b) of Figure S8 below) observed in Malinka et al. (2018) does not show this pattern in 730–950 nm. In addition, the spectral albedo of frozen ponds with snow (panel (c) of Figure S8

below) still shows the pattern, which is similar to yet less pronounced than that of snow pond and unponded ice. The simulation of Malinka et al. (2016) based on the radiative transfer theories of snow and white ice also shows the similar pattern.

[Figure]

**Figure 12.** Three special cases: thin wind crust on top of fine fresh snow of 4 cm thickness (left panels), a frozen-over gray melt pond with snow on top (middle panels), and a frozen-over crack with air bubbles and algae inclusions (right panels). The average air temperature is +0.3 °C (right panels) and −1.6 °C (the others).

**Figure S7. Spectral albedo of unponded sea ice and frozen ponds observed in Malinka et al. (2016).**

[Figure]

**Figure 7.** Frozen blue ponds. Polarstern-2012, Stations 1 **(a)** and 3 **(b, c)**. The left pond is heterogeneous. The sensor was placed approximately in the center of the photograph, about 1 m from the pond edge.

**Figure S8. Spectral albedo of frozen ponds observed in Malinka et al. (2018).**

- Based on the results mentioned above, we added statements about the agreement between this study and previous studies in the manuscript: "The result which is also consistent with observations in Malinka et al. (2016), which reported a similar nonlinear pattern shown both in the observed albedo of snow-covered ponded or unponded ice and in the simulated albedo based on radiative transfer theories." **(Line 296–298)** (Line 326–328 in mark-up version)

- **Line 288:** Can you better introduce $\alpha_{412}/\alpha_{667}$ as no results are presented in section 3.2 about $\alpha_{412}/\alpha_{667}$ and it is only in section 4 that references are made to this ratio.
- Thank you for point out this issue. The statement is intended to emphasize the limitation of the albedo ratios caused by observation time in this study. We changed this sentence and removed the name of certain ratio such as $\alpha_{360}/\alpha_{490}$ or $\alpha_{412}/\alpha_{667}$ to avoid the confusion: "It should also be noted that the albedo ratio in this study is developed based on...". **(Line 308)** (Line 338 in mark-up version)

- **Line 290:** "that some uncertainty remains in this result" Can you be more accurate about the uncertainties? Can the uncertainty be quantifiable?
- Thank you for the comment. The major uncertainties include volume of liquid water trapped under the ice lid (Flocco et al., 2015) and the thickness of snow cover (Anhaus et al., 2021), which are reported relevant to the albedo of snow pond, thus may cause significant change on spectral distribution. In addition, minor uncertainties such as thickness of ice lid or underlying ice have effect on albedo but do not cause great change for snow pond. Those uncertainties are hard to quantify since few snow-covered melt ponds were observed and reported in previous studies, especially in the Pacific sector of the Arctic. And we will focus on the albedo measurement of snow-covered melt ponds in the future expeditions to expand the dataset for a more robust conclusion. Combing with the constructive comment from the Editor, we have added discussions about the uncertainties including snow impurities, snow thickness and pond water in Section 3.2. **(Line 311–319)** (Line 340–351 in mark-up version)

- **Line 299:** "a correlation coefficient of 0.12 is found between" how is it calculated? Using values from figure 7 and figure 9?
- The correlation coefficient was calculated using the albedo and depth data of 50 melt ponds (of which the depth is shown in Figure 9)—out of a total of 81 ponds (of which the albedo is shown in Figure 7)—that the depth measurements were conducted during observation.

- **Line 333:** "a radiative transfer model" add "described in Section 2.2.
- We added the words in accordance with the comment. **(Line 361)** (Line 393 in mark-up version)

- **Line 394:** "except for the influence of temperature and radiation which is discussed in section 3.2", the influence should be reminded to complement the discussion.

- We added the influence of temperature and radiation: "…except for the influence of temperature and radiation which affects the energy budget in thermal processes, causes such as precipitation and wind also have effects on the formation of them". **(Line 425–426)** (Line 457–458 in mark-up version)

Section 4 discussions:

Wavelengths corresponding to MODIS bands are selected to identify the limits of the ratio used to identify snow covered pond. The analysis would be stronger if some in situ observations were compared with collocated MODIS measurements to assess how effectively the satellite performs and under which conditions MODIS is too limited. If a case study cannot be conducted, some references to MODIS pond identification should be cited and compared with the present study.

Two melt pond retrieval algorithms developed for satellite data are applied to the observations from the present study. Although references for both algorithms are provided, it would be helpful to introduce their underlying principles and to specify if any adaptations made to use them with in situ measurements. This could be included in the Supplementary Information to complement the study's methodology. Again, a case study comparing observations and Sentinel-2 data would be a valuable addition to extend the analysis to satellite observations.

Thank you for your constructive comment. We are regretful but it is not feasible to match or effectively compare in-situ data with satellite data due to the significant scale difference between the spatial resolution of MODIS (>250 m) and individual melt ponds (3–5 m). Besides, the Multispectral Imager (MSI) aboard Sentinel-2 has spatial resolutions from 10 m to 60 m which are closer to individual ponds, but we failed to found any image that overlaps with the snow ponds observed in this study (numbered as IC1202-2, IC1202-3, IC1204-2, IC1206-2, IC1804-1, IC1804-2, the last two of which provide spectral albedo shown in Figure 8 and Figure 13).

Therefore, case studies cannot be conducted based on current in-situ dataset and satellite product. But we will strive to compare observations and satellite data with the support of future expeditions, so as to better elaborate on the limitations of satellite observations.

To further illustrate the limitations of existing algorithms, we cited literature related to melt pond identification algorithms and added an introduction to their underlying principles before the citing two MPF algorithms: "The albedo ratio is also widely used in MPF (melt pond fraction) retrieval algorithms, which focus on deriving MPF from satellite data (Markus et al., 2002). For most algorithms, the albedo in certain bands measured by satellite sensors is operated to obtain a specific ratio, based on which

the clusters of snow/ice, melt ponds, and open water in scatter plot are determined. Albedo of certain area can be then converted to MPF based on its relative position in the plot". **(Line 449–453)** (Line 481–485 in mark-up version)

Based on the underlying principles, we analyzed the reasons why current algorithms fail to identify snow ponds as well as the potential improvement directions: "The reason behind this misidentification is surface state which causes different spectral characteristics of refreezing ponds from those 'typical' melt ponds. According to Figure 8, the difference between the albedo maximum (450-550 nm) and the low albedo in near-infrared (700-900 nm) reduces as the pond freezes, while most algorithms rely on those bands to distinguish ponds from unponded ice". **(Line 460–463)** (Line 493–495 in mark-up version)

Besides, although effective comparison between satellite data and in-situ data cannot be achieved with MODIS/Sentinel-2 products, we believe that as the resolution of satellite observation further improves, more detailed identification of melt ponds and even their types will become possible. The results of this study may provide theoretical support for the development of more advanced algorithms at that time.

Section 5 Summary:

The summary should be expanded into a forward-looking conclusion that clearly states the study's limitations and outlines concrete next steps. What is still required to improve the analysis and reduce the uncertainties? What would be necessary if these observations were conducted again (e.g., meteorological data and snow depth, etc.)? How will the parametrization of the radiative transfer model be used? Is there a potential study to improve the identification of snow-covered ponds from satellite? What kind of refreezing melt pond studies does the community need to advance understanding of sea ice albedo?

Thank you for your constructive comments. We expanded the summary and added statements about the limitations and outlooks of this study.

Although observation data during 5 Arctic expeditions were used, this study is still limited by the short of sample numbers and incomplete measurements. To improve the analysis and reduce the uncertainties, the following measures are needed: recording pond types in different dates to expand the dataset of proportion of surface states; comprehensively measuring parameters such as snow depth, ice lid thickness, pond depth, and thickness of underlying ice for all ponds; documenting local weather conditions in the preceding days of pond observation to support subsequent analysis; and investigating the potential use of albedo ratios in enhancing the retrieval of melt pond fraction from satellite data. Besides, since the characteristics of typical melt ponds are well documented and studied, we argue that the process during refreezing

period should be focused to further understand the albedo of Arctic sea ice.

The added sentences to the summary are as follows: **(Line 484–491)** (Line 520–527 in mark-up version)

"The melt pond dataset collected from five Arctic expeditions was used in this study, but we were still limited by the incomplete measurement in some stations and the short of sample numbers for several types of ponds, especially the snow pond. Hence the uncertainties of pond albedo (i.e. weather condition, snow depth and pond water) remain unclear, requiring detailed records on meteorological and physical properties to enhance current results. Moreover, the pond classification and the modified parameterization can be adopted to large scale sea ice mode (i.e. CICE) to improve the evaluation or prediction during refreezing period in the Arctic. The albedo ratios as indicators of snow pond or unponded ice provide insight on developing MPF retrieval algorithms with advanced identification of melt ponds. The radiative energy balance of refreezing melt ponds should be focused along with enhancive studies to further understand the Arctic sea ice."

**Technical comments**

- **Line 92:** the equations should be numbered.
- We numbered the equations in the manuscript. **(Line 101–121)** (Line 111–137 in mark-up version)

- **Line 317:** More consistency should be applied for defining the acronyms to avoid confusion. Some acronyms are defined in Figure 10 legend and then used in the text without explanation: line 312 "the fitting result is close to that of ML96 and SP07". Etc.
- Thank you for the comment. We revised this part accordingly. In the revised version, the acronyms are only used in Figure 10 to avoid mess caused by excessive annotations. The previous studies in the main text are all presented in the standard citation format. **(Line 339–341, 352)** (Line 371–374, 384–385 in mark-up version)

- **Line 416:** the opposite is also observed, PCA algorithm and LinearPolar Algorithm are defined in the text line 418 but not in the Figure 13 legend.
- We added citations in the caption of Figure 13. **(see the caption of Figure 13)**

- **Figure 11:** unit of H should be defined.

- We added definition of the unit of H in the caption of Figure 11. **(see the caption of Figure 11)**

- **Figure 13:** use color-coded makers as in previous figures to help readers distinguish between the different types of melt ponds.
- We added color-coded markers in Figure 8b and Figure 13 to help distinguishing different types of ponds. **(see Figure 8 and Figure 13)**

- **References:** Rösel and Kaleschke, 2011 is missing in the bibliography.
- Thank you for the comment. We apologize for the missing and the citation is now added into the bibliography. **(Line 612)** (Line 650 in mark-up version)

- **References:** There are two references for Wang et al. (2020). They should be labeled a and b to avoid confusion, or differently to distinguish the different authors.
- Thank you for point this out. We labeled a and b to distinguish the two references. **(Line 635,638)** (Line 673,676 in mark-up version)

**The list of all changes we made to the manuscript:**

Major changes:

1) We added discussions about the uncertainties of our result in Section 3.2, and emphasized the importance of uncertainties including snow impurity in Section 5.
2) The abstract is rephrased to clarify the goals of this study and to add more specific information of the observation.
3) We added more explanation about the surface classification and the potential use of this study.
4) We added some sentences in the introduction to explain the importance of refreezing period
5) We rewrote Section 2.2 to avoid simple repetition of Lu et al. (2016) and to emphasize the modifications we made to the model.
6) We added statements about the agreement between this study and previous studies in Section 3.2.
7) We cited literature related to melt pond identification algorithms and added an introduction to their underlying principles to better illustrate the limitations of existing algorithms
8) We added analysis on reasons why current algorithms fail to identify snow ponds and the potential way of improvement.
9) We expanded the summary into a conclusion by adding statements about the limitations and outlooks of this study.

Minor changes

1) A sentence about the importance of albedo in the introduction has been changed to avoid overstatement.
2) A brief definition has been added to help understanding the term "surface scattering layer".
3) The detail of measurement on physical properties such as ice lid thick or ponds depth is added in Section 2.
4) We added information of instrument and the corresponding wavelength in the caption of Table 1 to clarify the distinction between different radiometers.
5) We removed improper speculation about the annual reduction.
6) We increased the font size of Figure 6a.
7) We added a statement to clarify that the percentages of different type of melt pond in Section 3.2 are based on the number of observations.
8) We reorganized part of the sentences accordingly in Section 3.1 to improve the

readability.

9) We adjusted a statement in which description of ice type is removed to avoid confusion.

10) We rewrite the statement to better describe the source of noise.

11) An ambiguous description about ponds with surface partially or completely frozen is corrected to clarify its intended meaning.

12) We mentioned and emphasized the finding about identifying snow ponds in Section 5.

13) We added quantitative results about error from observations which is reduced our modification on model.

14) We added statement about comparison with observations in Amundsen Basin in Section 3.1.

15) A typo in Section 5 is corrected.

16) We rewrote a sentence about the effect of radiative forcing in the Arctic Ocean in the introduction to improve the accuracy.

17) The undefined lambda in equation 3 is now defined.

18) The equation and figure adapted from Lu et al. (2016) are acknowledged.

19) We rewrote the sentences with useless variables to clarify the features of the model and avoid confusion they may cause.

20) We added values of deviation between two instruments before and after calibration in Figure 5.

21) The missing explanations about Figure 6b are now added in the figure caption.

22) We added words to better describe the model used in Section 3.4 in accordance with the comment.

23) We added the influence of temperature and radiation in a statement to remind and complement the discussion.

24) We numbered the equations in the manuscript.

25) The previous studies in the main text of Section 3.3 are now all presented in the standard citation format to avoid the inconvenience of acronyms.

26) We added the citation of algorithms in the caption of Figure 13.

27) The definition of the unit of H in Figure 11 is added in the caption.

28) The color-coded markers are added in Figure 8b and Figure 13 to help distinguishing different types of ponds.

29) The missing citation and newly cited references are added into the bibliography.

30) We labeled a and b to distinguish two references.